

# Ensemble reconstruction of spatio-temporal extreme low-flow events in France since 1871

Laurie Caillouet[1], Jean-Philippe Vidal[1], Eric Sauquet[1], Alexandre Devers[1], and Benjamin Graff[2]

[1]Irstea, UR HHLY, Hydrology and Hydraulics Research Unit, 5 rue de la Doua, BP32108, 69916 Villeurbanne Cedex, France
[2]Compagnie Nationale du Rhône (CNR), 2 rue André Bonin, 69004 Lyon, France

*Correspondence to:* Laurie Caillouet (laurie.caillouet@gmail.com)

**Abstract.**

The historical depth of streamflow observations is generally limited to the last 50 years even in data-rich countries like France. It therefore offers too small a sample of extreme low-flow events to properly explore the long-term evolution of their characteristics and associated impacts. In order to overcome this limit, this work first presents a daily 140-year ensemble recon-
structed streamflow dataset for a reference network of near-natural catchments in France. This dataset, called SCOPE Hydro (Spatially COherent Probabilistic Extended Hydrological dataset), is based on (1) a probabilistic precipitation, temperature and reference evapotranspiration downscaling of the Twentieth Century Reanalysis over France, called SCOPE Climate, and (2) a continuous hydrological modelling using SCOPE Climate as forcings over the whole period. This work then introduces tools for defining spatio-temporal extreme low-flow events. Extreme low-flow events are first locally defined through the Sequent
Peak Algorithm using a novel combination of a fixed threshold and a daily variable threshold. A dedicated spatial match-
ing procedure is then set up to identify spatio-temporal events across France. This procedure is furthermore adapted to the SCOPE Hydro 25-member ensemble in order to characterize in a probabilistic way unrecorded historical events at the national scale. For the first time, extreme low-flow events are described and compared in a spatially and temporally homogeneous way over 140 years on a large set of catchments. Initial results bring forward well-known recent events like 1976 or 1989-1990,
but also older and relatively forgotten ones like the 1878 and 1893 events. These results contribute to improve our knowledge on historical events and provide a selection of benchmark events for climate change adaptation purposes. This study moreover allows for further detailed analyses of the effect of climate variability and anthropogenic climate change on low-flow hydrology at the scale of France.

## 1   Introduction

Hydroclimate projections for the 21st century generally agree on an increase in low-flow severity in France (see e.g. Chauveau et al., 2013; Vidal et al., 2015) – and more generally in Southern Europe (see e.g. Forzieri et al., 2014; Prudhomme et al., 2014; Giuntoli et al., 2015) – that could undermine current water management practice and require drastic measures for adapting water uses and for sharing resources among different economic sectors (irrigation, hydropower production, etc.). In this context of adaptation to climate change, a deep knowledge of major historical events that affected France constitutes a reference basis





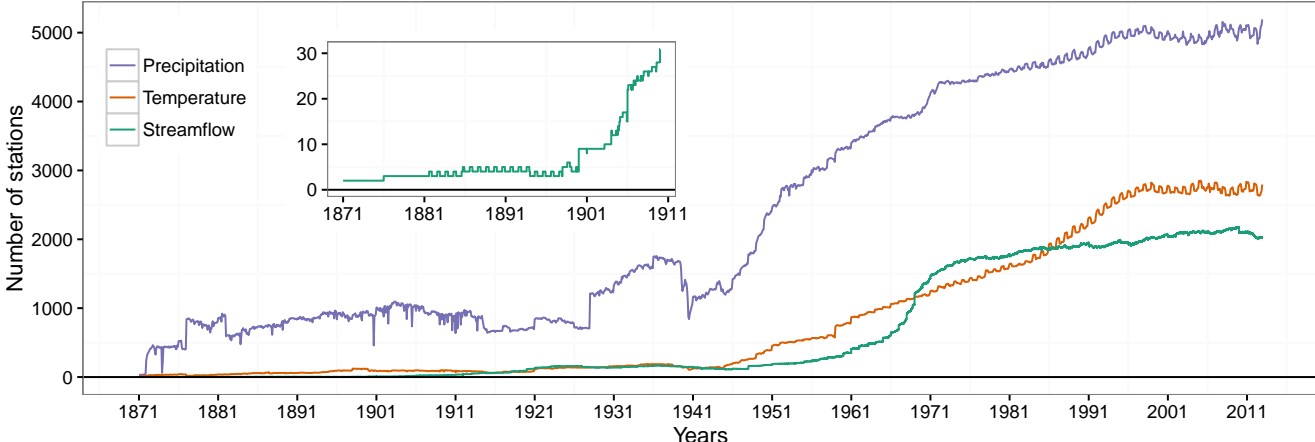

**Figure 1.** Evolution of the monthly averaged number of available precipitation and temperature stations in the Météo-France database (as of March 2015) since 1871 as well as the daily number of discharge stations in the Banque HYDRO French database since 1871.

for anticipating future severe low-flow events. Studying these events requires a dense network of long and reliable streamflow series to derive robust characterizations at the national scale. However, even in a data-rich country like France, few local observations are currently available in databases before the 1950s, as shown by Fig. 1. Data from less than 10 hydrometric stations were available before 1900, and their number shows a slow increase after 1950 only. Consequently, country-scale low-

flow studies using observations usually focus on recent periods when enough data are available (see e.g. Hannaford and Marsh, 2006; Giuntoli et al., 2013) or on few stations available on longer periods (see e.g. Hisdal et al., 2001; Pfister et al., 2006). Recent hydrometric data rescue actions for specific events (Auffray et al., 2011) or from specific historical sources (Le Gros et al., 2015) contributed to increase the breadth and depth of streamflow data over France, but they are still insufficient to allow studying extreme events at the national scale over long periods.

Reconstruction of streamflow time series through hydrological modelling, and using precipitation and temperature data as inputs, have been developed to overcome the limited historical depth of hydrometric records (see Jones et al., 2006, and references therein). Crooks and Kay (2015) and Lennard et al. (2015) for example reconstructed daily streamflow series for specific locations in the UK, using reconstituted historical meteorological data. Spraggs et al. (2015) extended meteorological time series back to 1798 to derive daily streamflow reconstructions and study the implications of extreme low-flow events for the

Anglian region in the UK. In France, comprehensive reconstructions of droughts and low flows from 1958 onwards have been carried out with a hydrometeorological modelling chain (Vidal et al., 2010b; Soubeyroux et al., 2010), providing a reference basis for future spatio-temporal drought projections (Vidal et al., 2012). The latter reconstructions were however limited by the availability of upper-air and surface meteorological data before the 1950s. Figure 1 illustrates this limited availability of surface temperature and precipitation observations in France. It has to be noted that climate proxies like dendrochronology

data may also provide relevant information for reconstructing streamflow with low – seasonal to annual – temporal resolution



(see e.g. Meko et al., 2012; Nicault et al., 2014). Dendrochronology data is most often used to derive meteorological drought indicators rather than hydrological drought indicators (see e.g. Cook et al., 2015; Labuhn et al., 2016).

A way to reconstruct meteorological data at high spatial and temporal scales – and then catchment-scale hydrological data – is to use climate downscaling methods from large-scale atmospheric and oceanic data. Auffray et al. (2011) for example recon-

structed hydrometeorological conditions that led to the 1859 flood of the Isère river (French Alps) based on purposely rescued pressure data and statistical downscaling with an analogue method. The recent release of two extended global reanalyses – the Twentieth Century Reanalysis (20CR, Compo et al., 2011) and the European Reanalysis of the Twentieth Century (ERA-20C, Poli et al., 2016) spanning the entire twentieth century (respectively from 1871 and 1900) – prompted studies aimed at deriving meteorological reconstructions on the 20th century through downscaling methods, and at using these to reconstruct stream-

flow series. Kuentz et al. (2013) reconstructed the 20th century hydrological variability of the Durance catchment (Southern French Alps) based on a analogue downscaling method (Kuentz et al., 2015) and a lumped conceptual hydrological model. More recently, Brigode et al. (2016) applied the same downscaling method and a hydrological model to reconstruct streamflow variability in a Northern Québec catchment. Few studies have been performed at a country scale, with the exception of Dayon (2015) who used a deterministic statistical downscaling approach combined to a physically-based hydrological model. Cail-

louet et al. (2016) recently provided a reconstruction of meteorological fields at a 8 km spatial resolution and daily temporal resolution in France through a probabilistic precipitation and temperature downscaling of 20CR.

When streamflow time series are available from observations or hydrological simulations, numerous methods allows defining and characterizing extreme low-flow events. Two main approaches can generally be distinguished (Tallaksen and Van Lanen, 2004). The first one considers low-flow characteristics like annual minimum n-day discharge (see, e.g. Smakhtin, 2001). The

second one focuses on characteristics of events temporally defined by deficits under a given threshold (see e.g. Fleig et al., 2006; van Huijgevoort et al., 2012; Van Loon and Van Lanen, 2012). The latter approach allows characterizing events at the local scale but has yet barely been used to study their spatial aspect. Some studies indeed looked at the areal extent of droughts as a spatial characteristic, but they did not identify independent drought events (e.g. Bonaccorso et al., 2013; Tallaksen and Stahl, 2014). Most of these studies focused on meteorological droughts and used a gridded and spatially homogeneous index to characterize

the spatial extent of drought events (often the Standardized Precipitation Index, McKee et al., 1993). Algorithms defining spatio-temporal events as a sequence of spatially contiguous and temporally continuous areas where the index is under a given threshold value have even been developed (Andreadis et al., 2005; Vidal et al., 2010b). However, to the authors' knowledge no study proposed a spatio-temporal definition of low-flow events based on station data, i.e. on data neither homogeneous nor continuous in space.

This paper proposes an ensemble reconstruction of spatio-temporal extreme low-flow events in France since 1871, through an ensemble downscaling of the 20CR extended reanalysis – based on the 20CR-SANDHY-SUB meteorological downscaled dataset developed by Caillouet et al. (2016) – and the subsequent hydrological modelling of a large number of near-natural catchments. Compared to previous low-flow studies, this study has three main combined specificities: (1) local events are characterized by a combination of a fixed threshold and a daily variable threshold, (2) they are defined in a probabilistic way



to take account of the uncertainties of the downscaling step in the streamflow reconstruction process, and (3) a spatio-temporal characterization of extreme low-flow events is provided at the scale of France.

The main objective of this work is to develop a method for identifying spatio-temporal extreme low-flow events based on probabilistic hydrometerological reconstructions. This work therefore presents beforehand the refinement steps applied to the 20CR-SANDHY-SUB dataset (Caillouet et al., 2016) to create the *SCOPE Climate* (Spatially COherent Probabilistic Extended Climate) dataset, and introduces the *SCOPE Hydro* (Spatially COherent Probabilistic Extended Hydrological) dataset, the hydrological dataset derived from SCOPE Climate. The last objective of this work is to present some examples of how characteristics of spatio-temporal extreme low-flow events may be analysed and exploited to extract relevant information from the SCOPE Hydro historical reconstructions running from 1871 onwards. The paper is structured as follows. Section 2 introduces the different reanalysis and observation datasets used. Section 3 describes the hydrological modelling step as well as the new spatial and probabilistic definition of extreme low-flow events. Section 4 shows some examples of reconstructed streamflow time series. Section 5 provides some characteristics of the extreme low-flow events that occurred in France since 1871. Results are finally discussed in Sect. 6.

## 2  Data

### 2.1  Observed streamflow

A set of 662 near-natural catchments in France has been selected for hydrological modelling and the study of extreme low-flow events (see Fig. 2). It is a combination of two existing datasets:

- The French low flow reference network consisting in 236 gauging stations, selected by Giuntoli et al. (2013) and meeting the following criteria: (a) at least 40 years of daily records, (b) the gauging station controls a catchment without direct human influence on river flow, (c) data quality is suitable for low flow analysis.

- A set of 632 gauging stations selected by Catalogne et al. (2014) and meeting the following criteria: (a) at least 26 years of daily records on the 1970-2005 period, (b) with no or few anthropogenic influence, (c) with a good quality during low-flow periods.

Streamflow data for these catchments were extracted from the French HYDRO database (http://hydro.eaufrance.fr/). Two case study catchments with contrasted regimes and long observation records will be used to exemplify local-scale reconstruction results in Sect. 3.1 and Sect. 5: the Corrèze@Brive-la-Gaillarde with an oceanic dominated regime, and the Ubaye@Barcelonnette, with a snowmelt dominated regime. Figure 18 in Annex A shows their daily interannual regimes. Observations are available since 1 January 1918 for the Corrèze and since 1 January 1904 for the Ubaye.




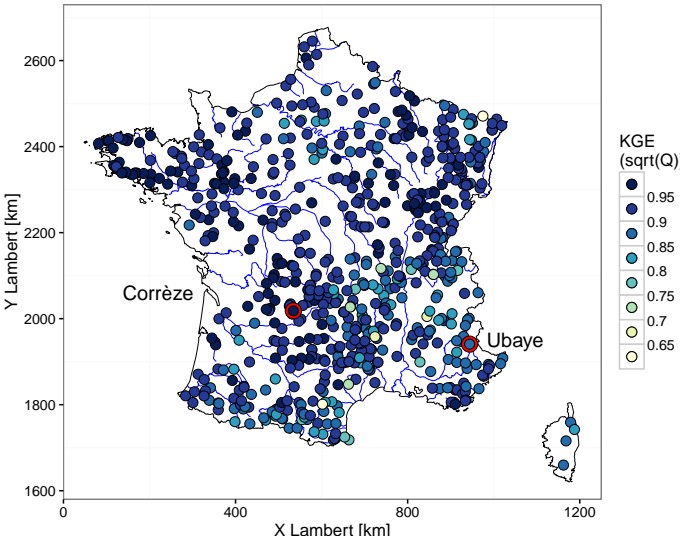

**Figure 2.** Location of the 662 hydrometric stations, and KGE calibration score for the hydrological model. Red circles highlight the two case study catchments: the Corrèze@Brive-la-Gaillarde (Western slope of the Massif Central mountain range) and the Ubaye@Barcelonnette (Southern Alps).

## 2.2 Meteorological forcing datasets

### 2.2.1 Safran near-surface reanalysis

Safran is the French near-surface meteorological reanalysis (Quintana-Seguí et al., 2008; Vidal et al., 2010a). It provides meteorological variables such as precipitation and temperature onto a 8-km resolution grid over France and at a daily temporal resolution from 1 August 1958 to present. Daily reference evapotranspiration is computed with the Penman-Monteith formulation (Allen et al., 1998). Safran data have been extensively used for hydrological studies, and notably for some of them dedicated to low flows (Vidal et al., 2010b; Soubeyroux et al., 2010; Pushpalatha et al., 2012; Nicolle et al., 2014). Catchment-scale data are computed with a weighted mean (for temperature) or sum (for precipitation and evapotranspiration) of each contributive cell of the 8-km grid to the catchment surface, based on the river network defined by Sauquet (2006).

### 2.2.2 SCOPE Climate

SCOPE Climate is a daily 8-km spatially coherent reconstruction of precipitation, temperature, and reference evapotranspiration fields since 1871 over France. It is based on the 20CR-SANDHY-SUB dataset that provides an ensemble of 25 analogue dates for each day during the period 1 January 1871 to 29 December 2012, independently for 608 climatically homogeneous zones paving France (Caillouet et al., 2016). These analogues are obtained through the statistical downscaling of the Twentieth





Century Reanalysis (20CR, Compo et al., 2011) with the SANDHY method (Stepwise Analogue Downscaling Method for HYdrology Ben Daoud et al., 2011; Radanovics et al., 2013; Ben Daoud et al., 2016) improved by two additional analogy levels (Caillouet et al., 2016). Analogues dates from the period 1 August 1958 to 31 July 2008 are converted to meteorological variables by resampling Safran reanalysis data.

The median of annual precipitation between Safran and 20CR-SANDHY-SUB for the 1959-2007 period shows a dry bias of around 10% on average over France (see top right panel of Fig. 7 in Caillouet et al., 2016). This bias, due to too many dry days resampled during the analogue selection, may call into question any interpretation of derived hydrological features – and especially extreme low flows –, and a bias correction step had to be set up. In a similar context, Timbal et al. (2006) for example introduced a correction factor to adjust the reconstructed rainfall series but this technique – as well as common bias correction

techniques – does not allow to retain the physical consistency and multivariate correlation structure inherent in the analogue approach. A resampling-based correction approach similar to the one adopted by Sippel et al. (2016) is therefore considered here: for each day between 1871 and 2012, the N analogues giving the lowest precipitation are removed. N analogues are then randomly resampled among the (25-N) left to keep a 25-member sample size. N, which is defined so that the bias with respect to Safran data is minimized, increases with the precipitation underestimation, with a maximum of 3 over France. This process

is independently done on each climatically homogeneous zone. Importantly, this resampling-based correction for precipitation does not affect the temperature bias and interannual correlation described in Caillouet et al. (2016) (not shown).

In order to build gridded time series from the 20CR-SANDHY-SUB dataset, Caillouet et al. (2016) independently combined analogues dates from one zone to another. The resulting lack of spatial continuity (see the discussion by Caillouet et al., 2016) could be heavily detrimental to the identification of spatio-temporal low-flow events. This issue is here addressed through the

Schaake Shuffle procedure, initially developed to reconstruct space-time variability in forecast meteorological fields (Clark et al., 2004). This procedure has been widely used as a post-processing of ensemble meteorological forecast fields for streamflow forecasting (see e.g. Robertson et al., 2013; Verkade et al., 2013; Demargne et al., 2014; Šípek and Daňhelka, 2015). Vrac and Friederichs (2015) also adapted it recently for multivariate bias correction of downscaled climate simulations. In the Schaake Shuffle approach, which can be seen as an empirical copula on rank correlation (Wilks, 2014), the ensemble members

are reordered so that their rank correlations across both space and variables match the ones from a randomly-picked sample of observed multivariate fields. In the present application, rank correlations are considered across the 608 climatically homogeneous zones and across the three variables, and observed fields are taken from the Safran reanalysis. For each target date, 25 dates are randomly selected within a 120-day window around the corresponding Julian day and among the period 1 August 1958 to 31 July 2008, a period consistent with the archive period for analogue dates in the SANDHY downscaling step

(Caillouet et al., 2016). Observed rank correlations are derived from the Safran multivariate meteorological fields from these dates and applied to the reconstructed ensemble, thus ensuring a spatial and inter-variable coherence of any single ensemble member.

The succession of steps (1) SANDHY (Radanovics et al., 2013; Ben Daoud et al., 2016), (2) subselection with additional analogue levels (Caillouet et al., 2016), (3) Bias Correction (see above), and (4) Schaake Shuffle (see above) is called SCOPE

and summarised by the 19 diagram in Annex B. The use of 20CR as the large-scale reanalysis input provides the SCOPE



Climate dataset. SCOPE Climate is available as a 25-member ensemble of Safran-like daily gridded time series of temperature, precipitation and reference evapotranspiration over the period 1 January 1871 to 29 December 2012. Catchment-scale data are computed as for the Safran data.

## 3 Methods

5    This section first presents the hydrological modelling step. The method specifically developed to characterize extreme low-flow events is then described in three steps: the local definition of events, the spatial matching of events and the application of this matching to the ensemble case.

### 3.1 Hydrological modelling

The hydrological model selected for this work is GR6J, a daily lumped continuous rainfall-runoff model developed specifically 10   for low-flows (Pushpalatha et al., 2011). It derives from the widely used GR4J model (Perrin et al., 2003) – which has been used recently in a reconstruction context by Brigode et al. (2016) – with a 5th parameter added to better model exchanges between surface and groundwater (Le Moine, 2008) and a 6th parameter added to better model low-flow periods (Pushpalatha et al., 2011). GR6J has been intensively used in France, in particular for low-flow studies (see Pushpalatha et al., 2012). It is here combined with Cemaneige (Valery, 2010; Valéry et al., 2014), a semi-distributed snow-accounting routine with 2 parameters.

15    GR6J is calibrated over the period 1 January 1973 to 30 September 2006 – called CAL in the following –, when more than 90% of the discharge stations have data, and which includes three well-known extreme low-flow events: 1976 (Bremond, 1976; Vivian, 1977; Zaidman et al., 2002, see e.g.), 1989-1990 (see e.g. Mérillon and Chaperon, 1990) and 2003 (see e.g. Moreau, 2004). A warm-up period of 3 years before the calibration period is considered to initialise the levels of the different water stores. The calibration uses Safran data as inputs and the Kling-Gupta-Efficiency (KGE, Gupta et al., 2009) on the square root 20   transformed streamflow (to give equal weights to high- and low-flows) as objective function. The KGE is computed as follows:

$$KGE = \sqrt{(r-1)^2 + (\alpha - 1)^2 + (\beta - 1)^2} \tag{1}$$

where $r$ is the linear correlation coefficient between simulation and observation, $\alpha$ the ratio between simulated and observed variance and $\beta$ the ratio between simulated and observed mean. The two Cemaneige parameters are calibrated only for stations 25   with a snowfall/rainfall ratio over 10% in order to prevent unrealistic values for stations without enough snow episodes. For these stations, the median values of the calibrated Cemaneige parameters are adopted. Calibrated models are then run with the two meteorological datasets described in Sect. 2.2 to obtain:

**Safran Hydro:** 662 daily streamflow series over the 1 August 1958 to 31 July 2014 period using Safran as input,

**SCOPE Hydro:** 662 x 25 daily streamflow series over the 1 January 1871 to 29 December 2012 period using SCOPE Climate 30      as input.





## 3.2 Characterizing extreme low-flow events

### 3.2.1 Local definition of extreme low-flow events

An approach based on deficit characteristics under a given threshold is adopted here to identify extreme low-flow events (Tallaksen and Van Lanen, 2004; Fleig et al., 2006). The first two panels of Fig. 3 present two commonly used thresholds.

A fixed threshold characterizes the low-flow season which depends on the hydrological regime of the catchment (see e.g. Tallaksen et al., 1997). A variable threshold characterizes any deviation from the normal seasonal pattern (see e.g. Van Loon and Van Lanen, 2012). A period of water deficit – filled in red Fig. 3 – is considered when discharge falls below the threshold level and continues so until the threshold is exceeded again. As the aim of the study is to characterize extreme low-flow events, the third panel proposes a mixed threshold as the daily minimum values of the two thresholds described above. This mixed

threshold thus allows identifying events deviating from the normal seasonal pattern only during the low-flow period.

The mixed threshold is computed here using the 90th percentile of daily streamflow over the CAL period: (1) from all days for the fixed threshold, and (2) independently from each Julian day for the daily variable threshold. The daily variable threshold is then smoothed through a 10-day moving average to improve the day-to-day consistency. Note that alternative daily variable threshold estimates using percentiles over moving windows around the Julian date may be equally suitable (Van Loon and

Laaha, 2015). The choice of percentile will highly condition the identification and characterization of extreme low-flow events. This aspect will be further discussed in Sect. 6.1.

The Sequent Peak Algorithm method (SPA, Vogel and Stedinger, 1987; Tallaksen et al., 1997) is then applied to pool extreme low-flow events. This method if often assimilated to a procedure for preliminary reservoir design. For a daily inflow to a reservoir $Q_i$ and a desired yield $Q_0$ corresponding to the mixed threshold defined above, the required storage at the beginning

of a period i, $S_i$, is defined by equation 2:

$$S_i = \begin{cases} S_{i-1} + Q_0 - Q_i, & \text{if positive} \\ 0, & \text{otherwise} \end{cases} \tag{2}$$

The fourth panel of Fig. 3 presents an example SPA curve, corresponding to the required storage $S_i$. An uninterrupted sequence of positive $S_i$ defines a period with storage depletion and a subsequent filling up. The event characteristics are then defined by a *severity* (required storage in the period, max(S), in mm), a *start date* (the beginning of the depletion period), an

*end date* (time of the maximum depletion), and a *duration* (number of days between these two dates). The bottom panel of Fig. 3 presents the formalisation of a single event used in Sect. 3.2.2 and Sect. 3.2.3 as a bar running between the start and end date.

The above procedure is applied for identifying and characterizing extreme low-flow events for each gauging station, independently on (1) observed time series, (2) Safran Hydro simulations, as well as (3) each of the 25 SCOPE Hydro reconstructions.



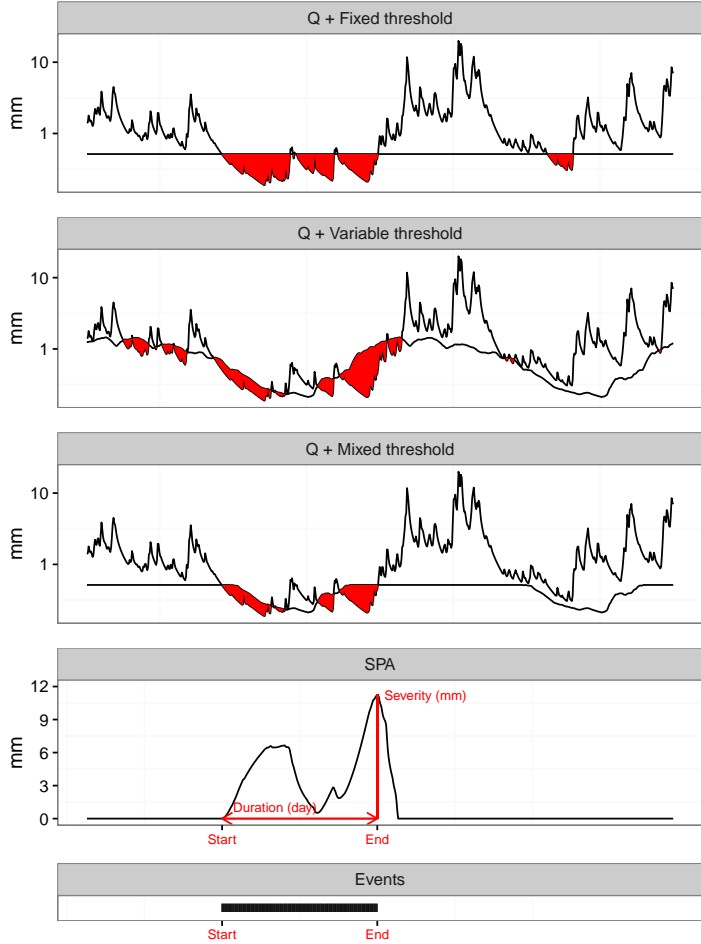

**Figure 3.** Extreme low-flow events characterization method. First panel: Fixed threshold. Second panel: Variable threshold. Third panel: Mixed threshold. Fourth panel: Sequent Peak Algorithm output using mixed threshold, and extreme low flow events characteristics. Fifth panel: Event definition (Start date to End date).



### 3.2.2 Spatial matching of extreme low-flow events

The development of a spatio-temporal event definition is required to precisely characterize a specific event across France. Indeed, events should be defined across the 662 stations to study for example the spatial extent of an event or the worst-affected region. The spatial definition developed in this section relies on a concept similar to the one developed by Uhlemann

et al. (2010) for assessing trans-basin flood events. The definition is presented below for a single set of independent events by station (deterministic case). The adaptation to the probabilistic case is developed in Sect. 3.2.3.

The spatial definition developed here only uses the start and end dates as represented in the bottom panel of Fig. 3. When an overlap of event dates occurs between different stations, the local events are considered as parts of the same spatio-temporal event. The spatial domain, i.e. the set of stations over which the matching is done, has next to be chosen. The matching is here

first done within French Hydro-ecoregions (HERs, Wasson et al., 2002). The zoning of France into 22 HERs shown in Fig. 20 of Annex C has been developed for the implementation of the EU Water Framework Directive. It is based on geology, relief and climate criteria to define biological, physico-chemical and hydromorphological reference conditions. The France subdivision in HERs have been widely used in recent hydrological studies (see, e.g., Sauquet and Catalogne, 2011; Cipriani et al., 2012; Snelder et al., 2013). The spatial matching is then done a second time across HERs to derive spatio-temporal extreme low-flow

events at the national scale.

This two-step approach originates in the large variability of extreme low-flow periods across catchments in France. Indeed, a single-step spatial matching gathering events from all 662 stations would lead to too much aggregated events in both space and time, possibly merging local events that do not proceed from a common meteorological driver (not shown). It may thus lead to spatially match an event occurring in the North-West of France with an event occurring in the South-East of France 6

months later, only because of the late recovery of a single North-West station and an outlier early start for a single South-East station.

Figure 21 in Annex D summarises the different steps described here. Figure 4 illustrates the whole spatial matching process for the 16 stations in HER 11 ("Causses aquitains", South-Western fringe of the Massif Central mountain range) during the 1 January 1989 to 31 December 1991 period. The corresponding procedure is detailed below:

1. Figure 4(a): Local events resulting from the SPA procedure are represented by black bars for each station.

     2. Figure 4(b) corresponds to results from the *within-HER matching* process in Fig. 21: Five spatio-temporal events built from temporally overlapping bars are identified here through different colours. This overlapping procedure may lead to pool formerly independent events for a given station within a single one: see for example Station 16 with the purple spatio-temporal event gathering several local events. In such a case, the new local characteristics of this extreme low-

flow event are computed from the former ones in the following way: the start date is the earlier start date, the end date is the latest end date, the duration is the sum of durations, and the severity is the maximum of severities. For each spatio-temporal event, a *HER-representative event* is defined using the median of start dates and end dates from all stations concerned. This HER-representative event is shown at the bottom of Fig. 4(b) with label "HER 11".





**Figure 4.** Spatial matching procedure for the local events reconstructed from Safran Hydro over the 1989-1991 period, through the example of the 16 stations in HER 11. (a) Independent events represented by black bars (see bottom of Fig. 3). (b) Matched events after within-HER matching. Each colour defines a spatio-temporal event at the HER scale. HER 11 representative events are represented at the bottom of the panel. (c) HER-representative events for the 22 HERs. Each black bar represents one event. (d) Matched HER-representative events after inter-HER matching. (e) Spatial HER-representative events matching reported to local events after France-wide matching. Each colour defines a spatio-temporal event at the France scale.




3. Figure 4(c): HER-representative events are shown for all 22 HERs with black bars. The HER 11 representative events obtained above are framed in red.

4. Figure 4(d) corresponds to results from the *inter-HER matching* process in Fig. 21: All HER-representative events are spatially matched together using the same overlapping process than in the (b) subfigure. The HER 11 representative events – again framed in red – are here pooled among only 3 distinct spatio-temporal events at the scale of France.

5. Figure 4(e) corresponds to results from the *France-wide matching* process in Fig. 21: The inter-HER matching for HER 11 representative events – bottom of Fig. 4(e) – is finally reported to each station through the *local allocation HER$_i$* process in Fig. 21, leading to some additional local pooling, and notably the merging of early 1988 events (red and grey in Fig. 4(b)) and of the late 1991 events (pink and brown in Fig. 4(b)). New local event characteristics – start and end dates, duration, severity – are computed using the same procedure as above.

Twenty catchments with major aquifers – most of them in HER 9 – for which extreme low-flow events can last several years are sidelined from the whole spatial matching procedure, including the building of HER-representative events. This prevents once again too much aggregation in both space and time. The France-wide definition of spatio-temporal events is reported to the local events for these stations by overlapping with their HER-representative events. This may therefore lead to local events in these catchments being identified as belonging to several distinct spatio-temporal events defined at the scale of France, and consequently to prevent any meaningful comparison of characteristics.

This spatio-temporal definition of events is applied to events from Safran Hydro between 1958 and 2012.

### 3.2.3 Adaptation to the ensemble case

Characterizing extreme low-flow events following Sect. 3.2.1 and applied to SCOPE Hydro leads to 25-member ensemble of local event definition for the 662 stations in France over the 1871-2012 period. The ensemble aspect leads to several questions: How many ensemble members out of 25 detecting an event are necessary to consider that it actually occurred? How to match events across the 25 ensemble members at the station scale? How to link spatially local ensembles of events that are detected, or not detected, or partially detected? This section attempts to provide responses to these questions by adapting the spatial matching described above to the ensemble case. The proposed approach goes along three steps: (1) reducing the ensemble reconstructions to a single deterministic series of extreme low-flow events to get back to the simpler Safran Hydro case, (2) applying the spatial matching described above to the resulting deterministic series, and (3) reporting the spatio-temporal definition of events to each ensemble member.

The overall idea is to derive a single deterministic series of low-flow events from the ensemble that matches approximately the behaviour of the one derived from Safran Hydro, and that matches more specifically the total time in extreme low-flow conditions given by the Safran Hydro simulation. The reduction to a single deterministic series is done independently for each station following the procedure below and as exemplified in Fig. 5 and Fig. 6 for Station 1 in HER 11.





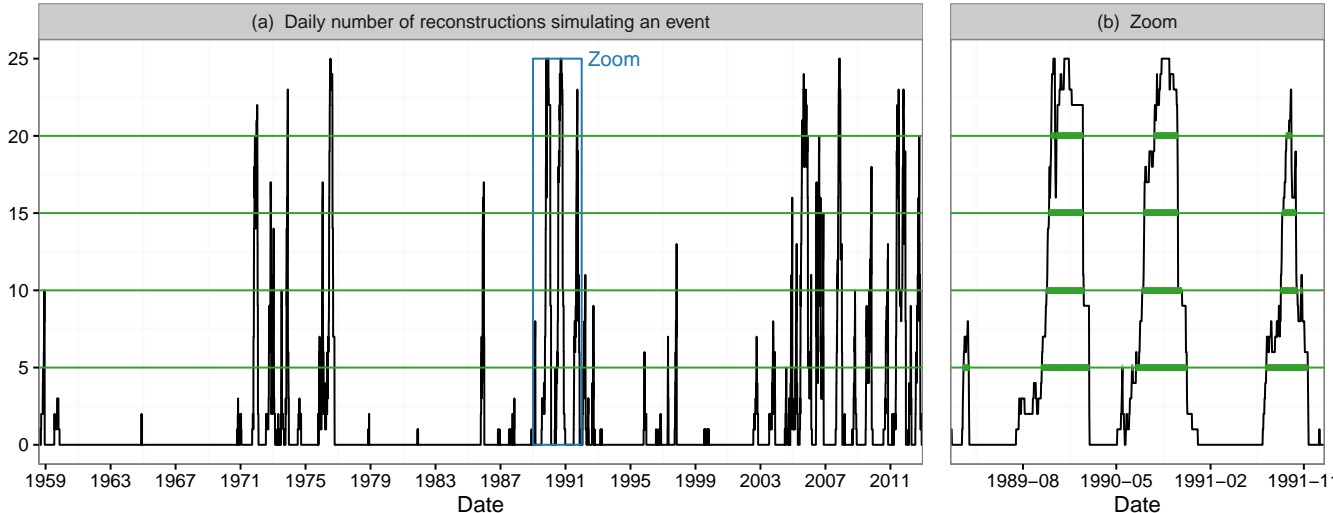

**Figure 5.** Deriving average events for Station 1 in HER 11. (a): Daily number of members simulating an event between the 1 August 1958 and the 29 December 2012. (b): Zoom on the 1989-1991 period. Examples are taken with 4 possible minima, set at 5, 10, 15 or 20 members.

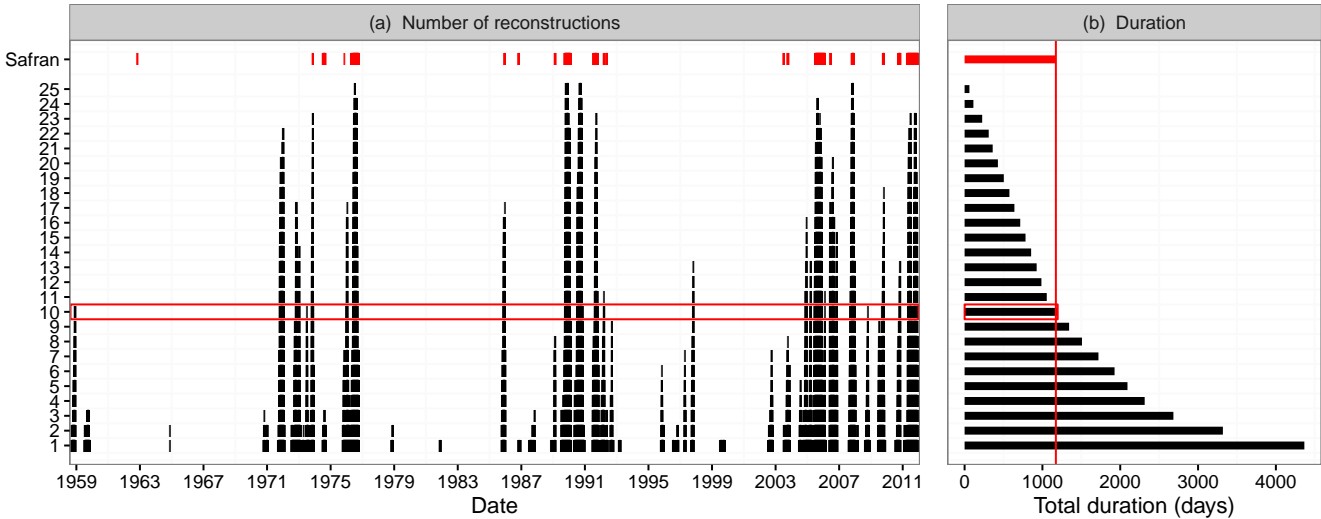

**Figure 6.** Choice of the minimum number of members for creating the average event series for Station 1 in HER 11. (a): Events from Safran Hydro and average series created with a threshold between 1 and 25 members. (b): Computation of the number of days simulating an extreme low-flow event for Safran Hydro and the 25 series of average events. The number of members providing the closest average events total duration to the Safran Hydro events total duration – here 10 – is selected.



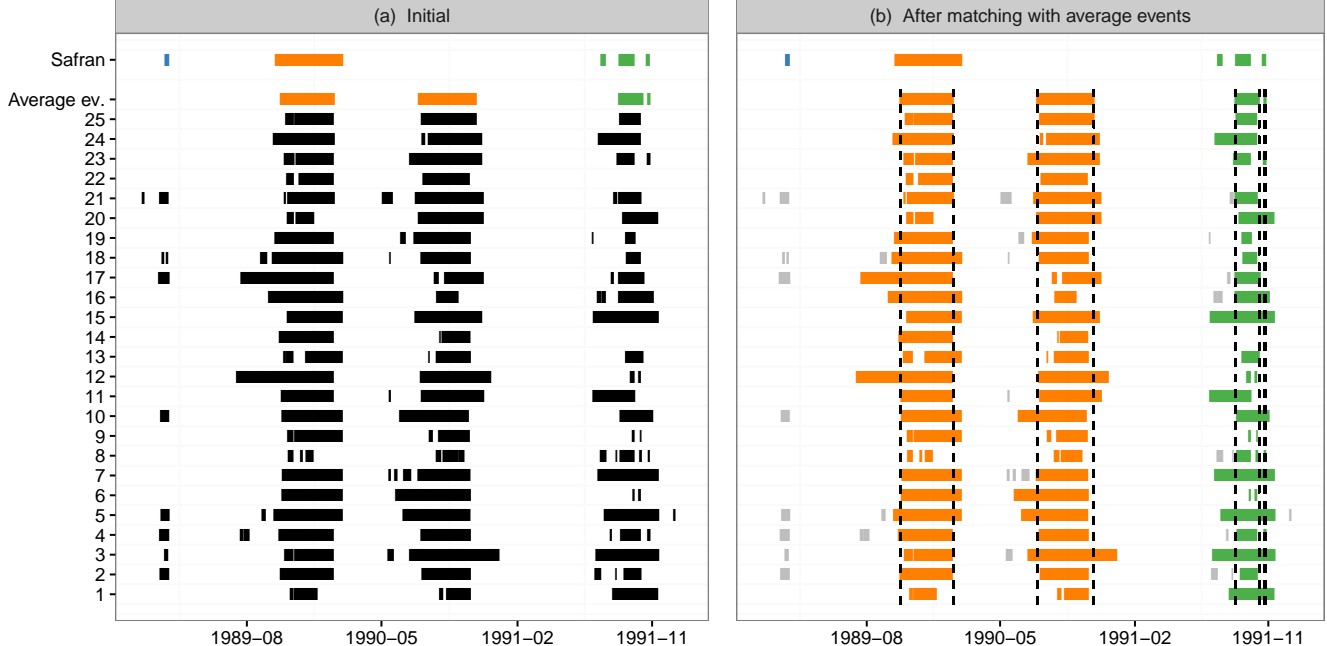

**Figure 7.** Ensemble spatial matching for Station 1 in HER 11 during the 1989-1991 period. Events from Safran Hydro, average event series and the 25 members of SCOPE Hydro are represented. (a): Each color defines one common spatial event for events from Safran Hydro and average series. Black bars are independent local events from the 25 members of SCOPE Hydro. (b): Matching between the 25 event members from SCOPE Hydro and average events. Black events not overlapping average events are considered as probabilistic noise and are removed from the dataset. Dashed lines are set at the start and end dates of each average event.

1. Figure 5(a): The daily number of ensemble members detecting an extreme low-flow event between 1 August 1958 and 29 December 2012 – the common period between Safran Hydro and SCOPE Hydro – is computed and represented by a curve (between 0 and 25 members). For a given number of members between 1 and 25, an *average event* is created each time this number intersects the curve. This event starts when the curve exceeds the number and the event continues until the curve falls below it. Fig. 5(a) plots 4 different possible number of members and associated average events.

2. Figure 5(b): For a number of members set at 5 (resp. 10, 15, 20), 6 (resp. 4, 3, 4) average events are created between 1989 and 1991.

3. Figure 6(a): The 25 possible series of average events are created between the 1 August 1958 and the 29 December 2012, using the 25 possible number of members. Actual events from Safran Hydro are also represented (in red).

4. Figure 6(b): The total duration of average events over the whole period is computed for each of the 25 series. The chosen number of members is the one giving the closest value to the total duration of Safran Hydro events over the same period (10 for this specific station).





The spatial matching developed in Sect. 3.2.2 is then applied to these average events considered as a single deterministic series of reconstructed events similar to this from Safran Hydro.

The final step deals with reporting spatio-temporal (deterministic) events to each ensemble member of SCOPE Hydro. It is illustrated in Fig. 7 with the same station as above over the 1989-1991 period, and goes as follows:

1. Figure 7(a): Events from all 25 SCOPE Hydro ensemble members are represented as well as average events. Safran Hydro events are also represented, but only for a discussion in Sect. 6.3. Considering a number of 10 ensemble members, four independent average events are available on this period (see also Fig. 5(b)). After the spatial matching, they are pooled into two spatio-temporal events (orange and green).

2. Figure 7(b): For each of the 25 ensemble members, events temporally overlapping with average events are matched together. Ensemble member events not overlapping with an average event (in grey) are removed. This actually concerns events detected by less than the chosen number of ensemble members (here 10).

The choices made during the spatial and probabilistic matching will be further discussed in Sect. 6.2 and 6.3.

Characteristics of extreme low-flow events – start date, end date, duration, and severity – are then computed independently for each ensemble member following the rules detailed in Sect. 3.2.2. A duration of 0 days and a severity of 0 mm are attributed to an ensemble member that does not detect an event, e.g. the ensemble member #22 for late 1991 event in Station 1 of HER 11 (see Fig. 7). This allows keeping an homogeneous ensemble definition of events, which is particularly relevant when considering ensemble statistics like the median value of extreme low-flow event characteristics (see Sect. 5).

The spatial matching procedure adapted to SCOPE Hydro finally leads to spatio-temporal extreme low-flow events that are characterized locally for each station by an 25-member ensemble of start date, end date, duration, and severity.

### 3.2.4 Conventions for representation purposes

Spatio-temporal events from SCOPE Hydro are assigned a specific *spatial centre date* for plotting corresponding local summary characteristics along a temporal dimension. For each station where at least one ensemble member detects the events, a local median start date is computed. The spatial start date is then computed as the median dates over all stations detecting the event. The same procedure is applied for end dates, and the spatial centre date is taken as the middle date between the spatial start date and the spatial end date. It thus allows plotting characteristics – start and end dates, duration, and severity – of a specific spatio-temporal event reported locally for different stations in a homogeneous way along the temporal axis.

The daily number of stations concerned by a given spatio-temporal low-flow event can be computed. An estimate of the spatial extent of this event in terms of percentage of France surface area is computed based on the surface area of each HER and the percentage of stations hit within each HER weighted by catchment areas. This estimates is used to compute the maximum spatial extent reached during each spatio-temporal event.

Duration and severity may be expressed in terms of return period in order to compare event characteristics among themselves and across different catchments. As extreme low-flow events are not a regular phenomenon occurring every $T$ year, the approach developed by Shiau and Shen (2001) is used here: a mean interarrival time between two events is considered as the




basis for computing return periods. Severity and duration variables can further be equal to zero. This is accounted for in the following way. Let $X$ denote the target random variable and $F_X$ denote its cumulative distribution function. The cdf can be computed as follows using the total probability law:

$$\forall x > 0, F_X(x) = p_0 + G(X) \times (1 - p_0) \qquad (3)$$

5   where $p_0$ is the probability that $X = 0$ and is simply estimated as the empirical frequency of zeros in the observed sample, and $G(x)$ is the cdf of the distribution estimated on the non-zero values in the observed sample. A Generalized Pareto Distribution with 3-parameters (GPD3) for the duration and a Pearson type III (P3) distribution for the severity are selected based on a L-moment diagram (not shown). These distributions are calibrated for each reconstruction ensemble member and each catchment. Only ranges in return periods are considered in order not to put too much confidence in fitted values.

10  **4   SCOPE Hydro: Examples of reconstructed streamflow time series**

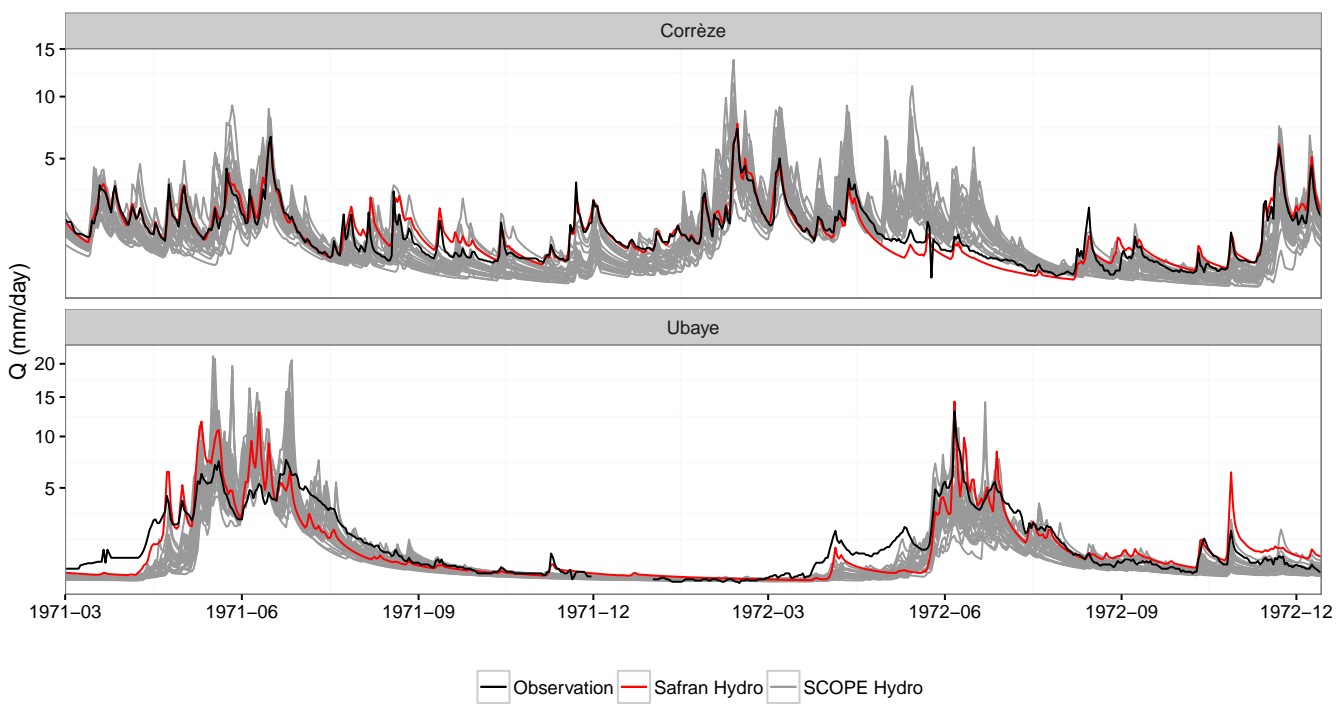

**Figure 8.** Corrèze and Ubaye daily discharge time series during the extreme low-flow event of 1972, from observations, Safran Hydro and SCOPE Hydro. Note the different root scales for the y-axis.





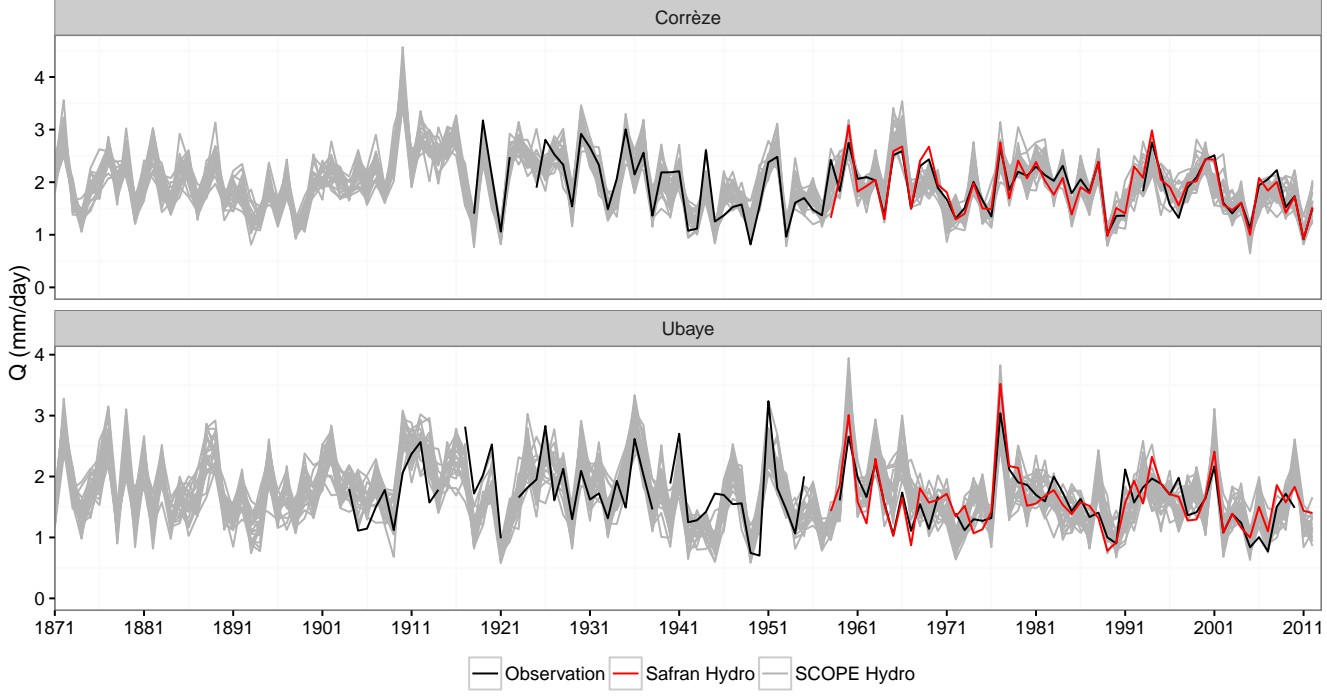

**Figure 9.** Corrèze and Ubaye annual discharge time series during the 1871-2012 period, from observations, Safran Hydro and SCOPE Hydro. Note the different scales for the y-axis.

This section briefly gives some quantitative results on the hydrological model calibration, and then shows SCOPE Hydro reconstructed streamflow time series for the two case-study catchments to briefly illustrate the hydrological inputs used to study extreme low-flow events.

Figure 2 shows the KGE calibration values for all 662 modelled catchments. More than 70% of the catchments have values over 0.9. KGE values are lower for mountainous areas (Massif Central, Alps, Pyrenees) and the minimum value is 0.61. It has to be noted that thorough validation experiments not shown here – out-of-sample experiments, split-sample experiments – have been performed to carefully quantify the overall hydrological modelling performance.

Figure 8 presents the observed and simulated daily streamflow time series for the Corrèze and Ubaye catchments during the extreme low-flow event of 1972 (see e.g. Duband, 2010; Chauveau et al., 2014). The temporal dynamics at both the seasonal and daily scales are fairly well simulated by SCOPE Hydro for both catchments. SCOPE Hydro but also Safran Hydro tend to underestimate spring streamflow for the Ubaye catchment for these specific years, possibly due to a non-satisfactory modelling of the early snowmelt in the hydrological model (see also Fig. 18). Safran Hydro and observations are most of the time included in SCOPE Hydro ensemble range, showing a good reliability of the ensemble reconstructions for this period. Additionally, the low-flow periods – Summer 1972 for the Corrèze and Winter 1971-1972 for the Ubaye – are reasonably well simulated.




Figure 9 shows the annual streamflow time series for the two catchments over the whole period. The observed interannual variability is well simulated by both Safran Hydro and SCOPE Hydro. Here again, Safran Hydro and observations are included in the SCOPE Hydro ensemble range. SCOPE Hydro allows to fill data gaps for years with missing observations (for example around 1925 for the Corrèze and 1918 for the Ubaye) and to extend our knowledge back until the end of the 19th century.

## 5   Extreme low-flow events since 1871

This section provides example results that can be derived from the SCOPE Hydro reconstruction dataset after the definition of extreme low-flow events (Sect. 3.2.1) and the ensemble and spatio-temporal matching (Sect. 3.2.2). The first part focuses on spatio-temporal events examined only at the local scale for the two catchment case studies (Corrèze and Ubaye). The second part then provides summary results on spatio-temporal events drawn at the scale of France.

### 5.1   Temporal ensemble reconstruction of extreme low-flow events for the Corrèze and Ubaye catchments

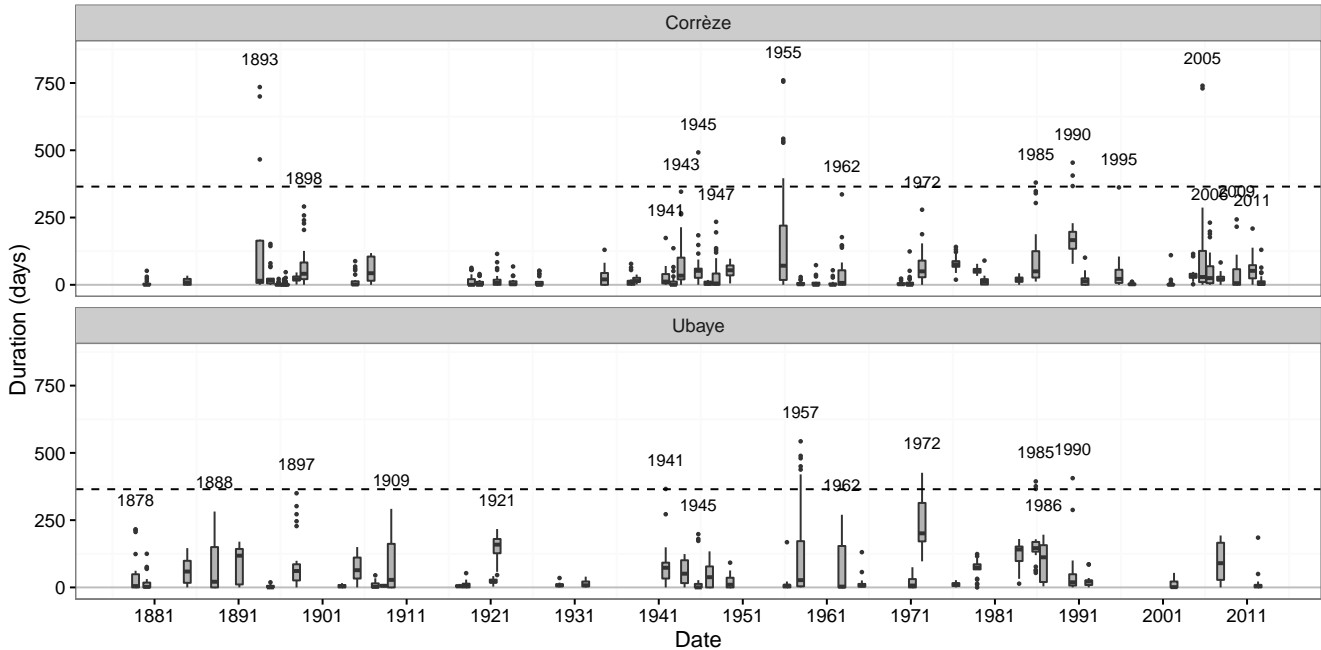

**Figure 10.** Extreme low-flow event duration (in days) for the Corrèze and Ubaye catchments over the 1871-2012 period. The x-axis represents the spatial center dates of the events (see Sect. 3.2.4). The 25 ensemble members are represented with a boxplot for each event. The lower and upper hinges correspond to the first and third quarters, respectively. Wiskers extend to 1.5 times the interquartile range. Events with remarkable values from any given ensemble member are labelled with the year of the spatial center date. The dashed line shows the 1-year duration for information.




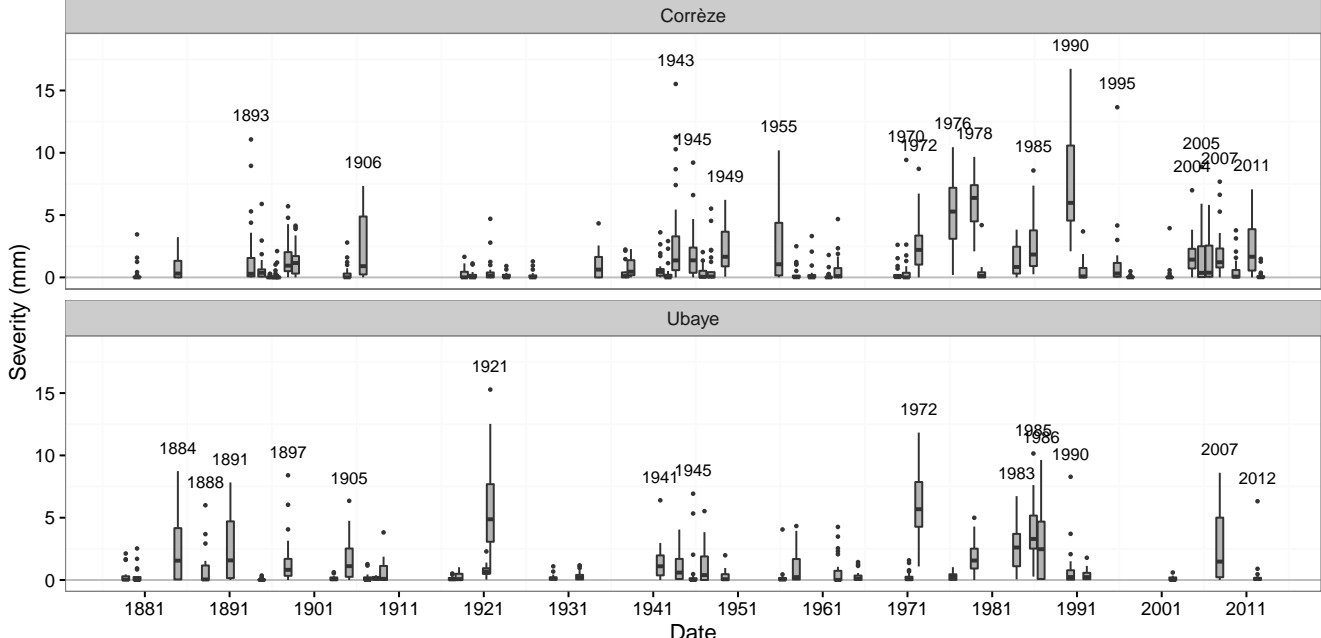

**Figure 11.** As for Fig. 10 but for severity (in mm).

Figure 10 shows the ensemble durations for each spatio-temporal extreme low-flow event as experienced by the Corrèze and Ubaye catchments over the 1871-2012 period. For both catchments some long events occurred at the end of the 19th century – for example 1893 and 1898 for the Corrèze, and 1878, 1888, and 1897 for the Ubaye – followed by a wet period until the beginning of the 1940s, interrupted only by the 1921 event. Several clusters of events can be identified: during the 1940s and 2000s for the Corrèze, and 1940s and 1980s for the Ubaye. Other long events also occurred in-between these clusters like the 1955, 1962 and 1972 events for the Corrèze, and the 1957, 1962 and 1972 events for the Ubaye. Even if long events are for a large part different from one catchment to another, some common ones may be identified in 1945, 1962, 1972 and 1990. The majority of events are not detected by all 25 ensemble members – as shown by the bottom of boxplots reaching 0 days – with the exception of some specific events like the 1990 event for the Corrèze and the 1921 and 1972 events for the Ubaye. This illustrates the large uncertainty in the characterization of events, as each of the 25 ensemble members is a potential reconstruction of the event.

Figure 11 shows in a similar way the severity obtained for each spatio-temporal extreme low-flow event for both catchment case studies. As for the duration, a wet period occurred between 1910 and 1940, here again interrupted by the 1921 event. Some events appear as exceptionally severe like the 1976, 1978 and 1990 events for the Corrèze, as well as the 1921 and 1972 events for the Ubaye. Moreover, these events are detected by all 25 ensemble members.

Based on Fig. 10 and Fig. 11, a higher number of events occurred for the Corrèze catchment than for the Ubaye catchment. A higher number of extreme events and higher severity values are simulated after 1940 for the Corrèze. These reconstructions





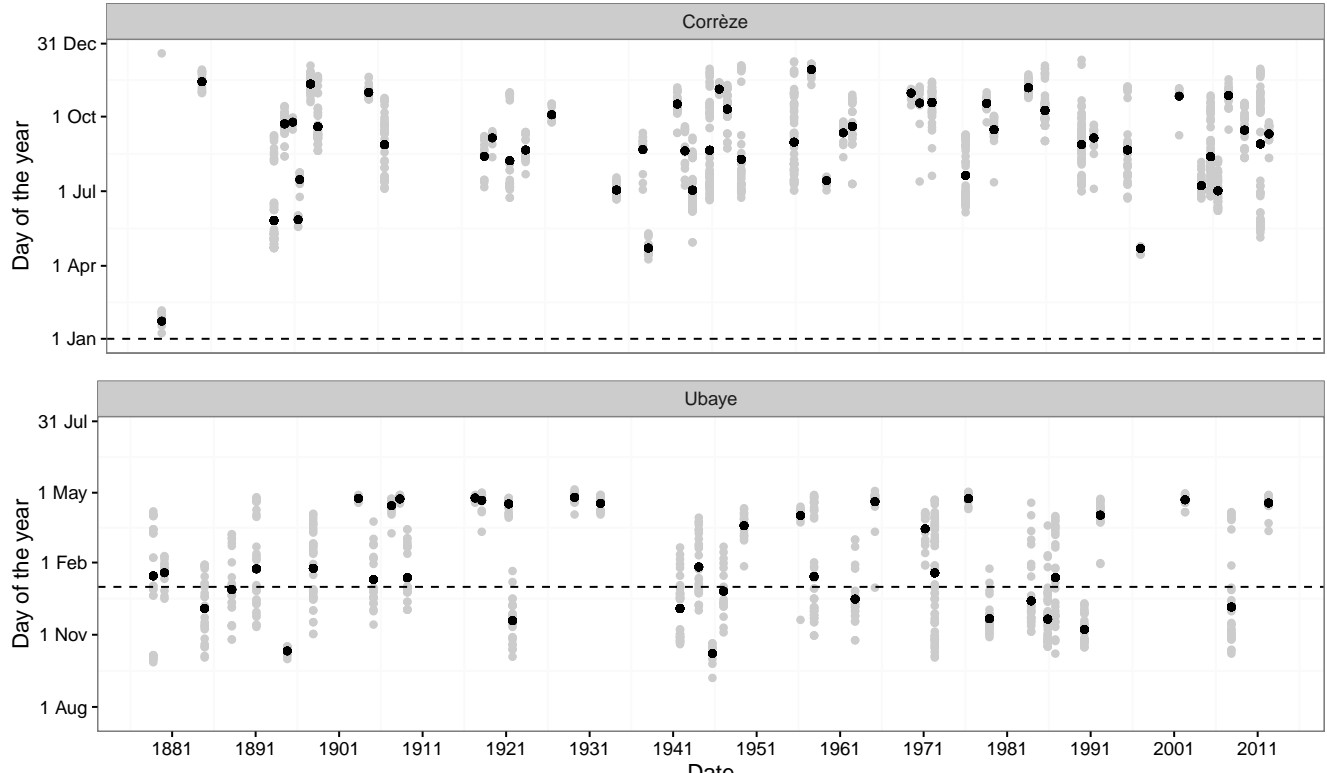

**Figure 12.** Extreme low-flow event start date (Julian days) for the Corrèze and Ubaye catchments over the 1871-2012 period. The x-axis represents the spatial center dates of the events (see Sect. 3.2.4). The dashed line is set on the first of January. Grey points represent the start dates of individual ensemble members detecting the event. Black points represent the average date of the grey points.

also identify long and/or severe low-flow events during periods for which no streamflow observation is available, notably the last decades of the 19th century.

Figure 12 presents the start date of each spatio-temporal extreme low-flow event for the Corrèze and Ubaye catchments over the 1871-2012 period. An average date representing the ensemble central tendency is computed using a seasonal index 5 (Burn, 1997). As for Fig. 10 and Fig. 11, there is a high variability among ensemble members, with start dates of individual members lying within a range of +/- 2 to 3 months around the average start date. The seasonality of the start dates illustrates the different regimes in the two catchments. Indeed, extreme low-flow events start between April and November for the oceanic-dominated regime (Corrèze), and between October to April for the snowmelt-dominated regime (Ubaye). The only exception is the particular event of 1880 which began in December 1879 for the Corrèze river. There is no visible trend on the seasonality 10 of start dates.

Figure 13 provides a two-way summary of spatio-temporal extreme low-flow events in terms of duration vs severity for both catchments, by plotting characteristics from individual ensemble members, but also as the median duration of the ensemble



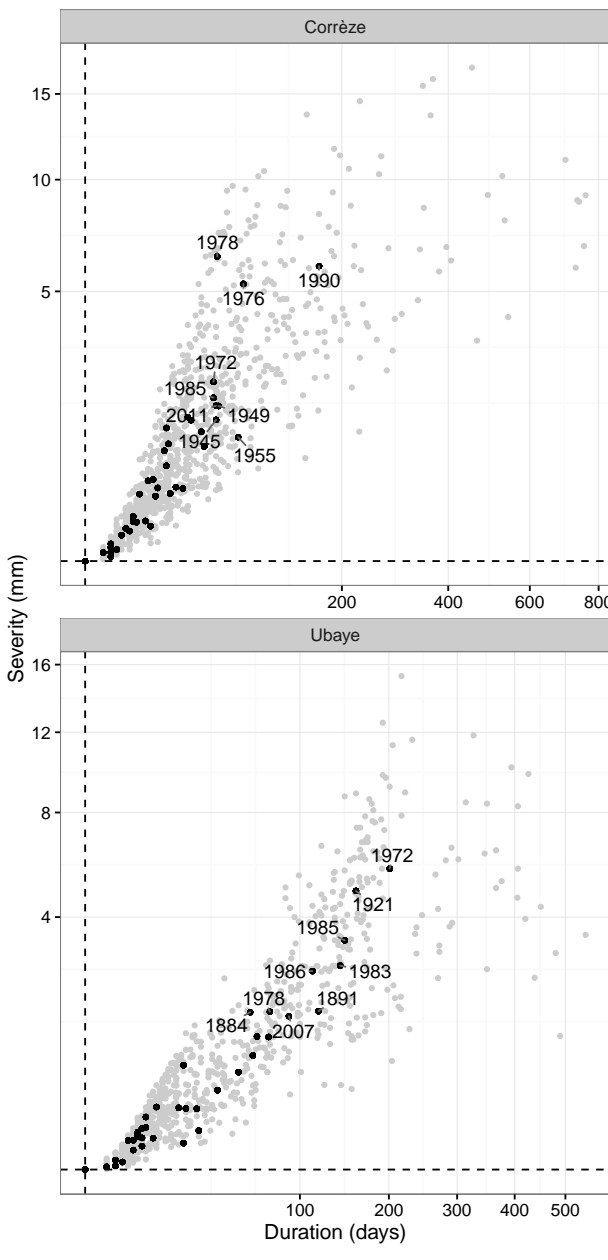

**Figure 13.** Duration of extreme low-flow events versus their severity for the Corrèze and Ubaye catchments over the 1871-2012 period. Grey points represents individual ensemble members. Black points represents the median characteristics of the 25-member ensembles. Events remarkable given their median duration or severity are labelled with the year of their spatial centre dates. Note the square root scale for both axes.





vs the median severity. These plots first show the unsurprisingly high correlation between these two characteristics. The 1976, 1978 and 1990 events for the Corrèze as well as the 1972 and 1921 events for the Ubaye appear to be exceptional over the 1871-2012 period. Figure 13 allows identifying reconstructed ensemble members with specific combinations of duration and severity, an information not provided by Fig. 10 and Fig. 11. Some ensemble members generate for example very long events,

5    but without exceptional severity values.

## 5.2   Spatio-temporal characterization of extreme low-flow events since 1871

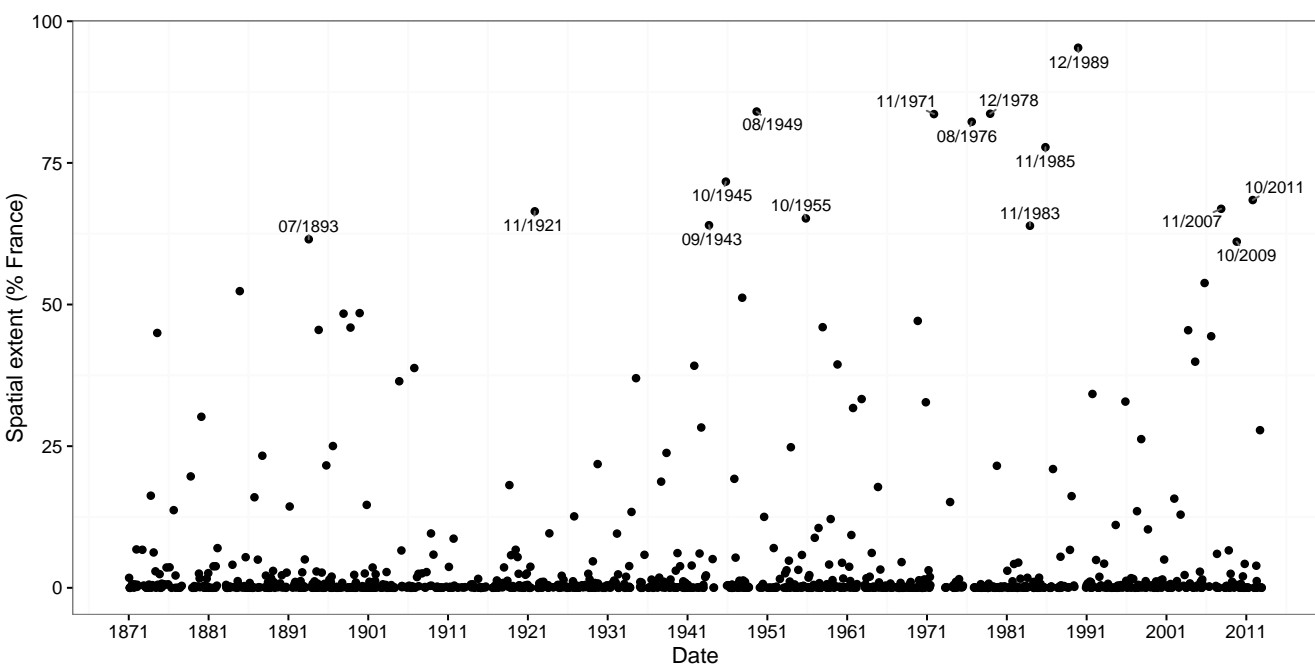

**Figure 14.** Median value of the maximum spatial extent of spatio-temporal events from individual ensemble members of SCOPE Hydro over the 1871-2012 period. See text for details. X-axis coordinates are the median of the dates when the maximum spatial extent is reached. Years and months of the median dates are labelled for events with a spatial extent over 60% of France.

The probabilistic and spatial matching done in Sect. 3.2.3 ensures the spatial consistency of extreme low-flow events. This critically allows studying these events at the national scale. Unless mentioned otherwise, plotted values in the following figures are median values over the 25 ensemble members.

10    Figure 14 introduces the spatial extent of the spatio-temporal extreme low-flow events identified over the 1871-2012 period. This spatial extent represents the maximum percentage of France hit by the event in terms of surface area, regardless of the intensity or duration of the event. Some events already mentioned in Sect. 5.1 are emphasized such as the 1893, 1921, 1945, 1976 or 1990 events. This last event is the only one having affected almost the whole of France. More generally, this figure highlights the fact that the only events having hit more than 70% of France occurred after 1940.



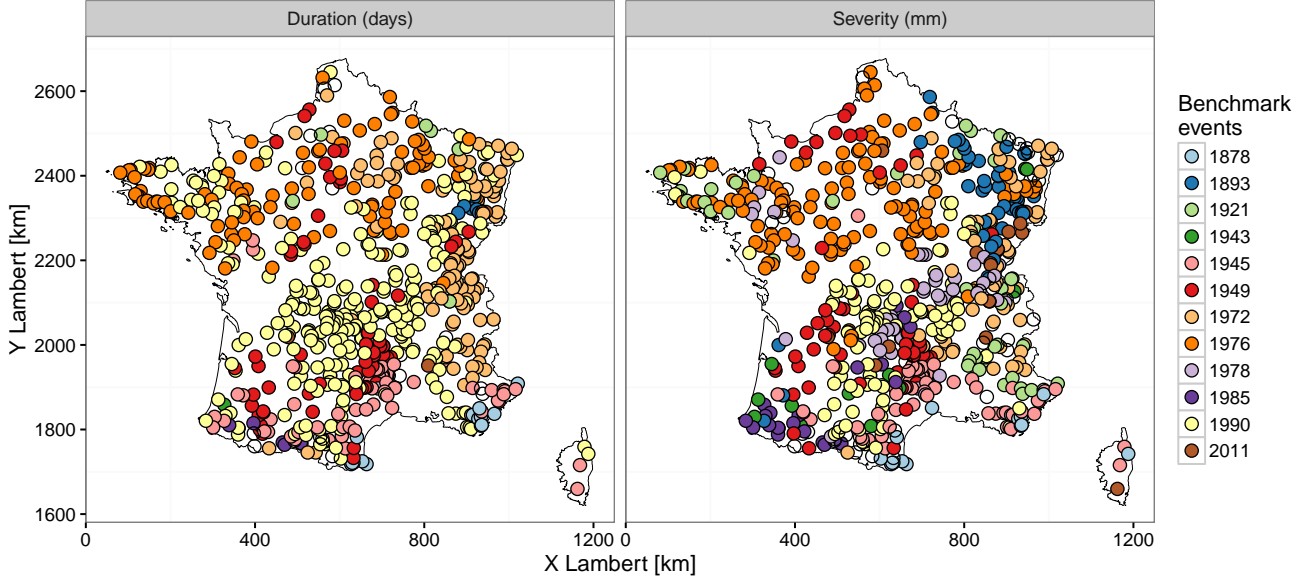

**Figure 15.** Benchmark events selected from median characteristics over the 1871-2012 period. Filled white circles identify stations which share their most extreme event with less than 4 other ones. These events are not listed in the legend.

Figure 15 highlights the longest and most severe events that hit individual catchments. Two events are highlighted in terms of duration: the 1976 event for the north of France and the 1990 event for the centre of France. The 1972 (resp. 1945) event was the longest one having hit the east (resp. south). The pattern is more patchy for the most severe events. The 1976 event is still predominant for the north of France. The 1878 event was the most severe in the north-east whereas the 1893 event still

5  has the record for some parts of the Mediterranean coastal area. The 1949 and 1990 events share the first place for the centre of France. This 1921 event is highlighted for different stations scattered in France, from the north-west to the south-east. It may have been an exceptionally severe event for the entire France even if not the most severe for all stations. Note that the 2003 event, which is often used as a reference at the European scale (see e.g. Laaha et al., 2016) is not highlighted as the most severe event for any station in France. These events are identified as *Benchmark events* and their characteristics can be drawn

10  for France as a whole.

Four of these benchmark events are further studied in Fig. 16. The 1878 and 1893 events have been selected because of the lack of related quantitative or even qualitative hydrometric observations available. Information on their meteorological drivers – or their socio-economic impacts – have however been documented elsewhere: see Moureaux (1880b) for the 1878 event, and Vimont (1893), Plumandon (1893), Dehérain (1893) or Cook et al. (2015) for the 1893 event. On the contrary, the 1976 and

15  1990 droughts have been largely studied in terms of low flows in France: see e.g. respectively Bremond (1976) and Mérillon and Chaperon (1990). The first two columns of Fig. 16 show the local return period of duration and severity of these spatio-





**Figure 16.** Median value of return period (in years) for duration (left) and severity (middle), and median of the number of days after the earliest start date (right) for the 1878, 1893, 1976 and 1990 extreme low-flow events. The earliest start date is shown on the top right corner for each event. Filled grey circles represent stations that did not detect the event.





**Figure 17.** Return period in severity for the 25 ensemble members of the 1893 event. Filled grey circles represents stations that did not detect the event.





temporal events. They show that the four selected events had very different spatial patterns. The 1878 event only hit the south of France, and particularly the Mediterranean coast. On the contrary, the 1976 only spared this coastal area. The 1893 and 1990 events both hit the entire France excepted the high-elevation snow-influenced alpine stations – as well as some lower elevation stations in the south for the 1893 event. Return periods for duration and severity highlight regions with the most extreme values

reached during each event: north-east for the 1893 event, and northern half of the country for the 1976 event. The 1990 event on the contrary was exceptionally long over most of France, as already suggested in Fig. 14. Differences between the two return periods may also be spotted: the 1893 and 1976 events were more outstanding in terms of severity – with return periods higher than 50 years for many stations – than duration, whereas the 1990 event was exceptional in terms of duration, with return periods higher than 20 years or even 50 years for the majority of stations. The third column presents the start date for

each of the four extreme low-flow events. Both the 1893 and 1990 events start dates lies within a common 6-month period for all stations concerned. On the contrary, two groups of stations are distinguished for the 1878 and 1976 events, with one of them displaying start dates occurring more than a year later than the first reconstructed one. This may be a consequence of choices made for the spatial matching and will be further discussed in Sect. 6.2.

All above results are drawn from median characteristics of the 25-member ensemble reconstruction. To give an idea of the

variability of spatio-temporal characteristics within this ensemble, Fig. 17 presents the 25 individual reconstructions of the 1893 event in terms of severity. Even if most of the members display an intense drought in North-Eastern France, some of them also simulate an extreme event in the centre (e.g. member 11 and 24), the west (e.g. member 1 and 21) or the south-west (e.g. member 4 and 8). All members however indicate that the Mediterranean coast was not severely hit by this event. This high variability among the ensemble members has already been noticed for the meteorological reconstructions (Caillouet

et al., 2016). The 25 maps in Fig. 17 represent 25 plausible reconstructions of the 1893 extreme low-flow event that may have occurred under the synoptic conditions from the 20CR reanalysis for this period. This aspect will be further discussed in Sect. 6.5.

## 6   Discussion

This section discusses some issues previously broached. The sensitivity of the definition of local extreme low-flow events

through the Sequent Peak Algorithm to the threshold is presented in Sect. 6.1. Different issues raised by the spatial matching and the ensemble matching are then discussed in Sect. 6.2 and Sect. 6.3, respectively. Section 6.4 touches upon issues that may arise when comparing spatio-temporal events from two datasets. Some limitations of summary statistics derived from the ensemble values is discussed in Sect. 6.5. Lastly, Sect. 6.6 discusses additional sources of uncertainties not considered in this study.

### 6.1   Threshold sensitivity for the local definition of events

This study is based on a threshold defined in Sect. 3.2.1 for identifying local extreme low-flow events. Changing the threshold – even within the 70-95th percentile range commonly used for defining low-flow thresholds (Hisdal et al., 2001; Fleig et al.,





2006; Van Loon and Van Lanen, 2012) – would lead to a different definition of events and different results in Sect. 5. A higher threshold value, e.g. based on the 80 percent exceedance probability (Q80), would for example lead to a higher number of events, but also to a new definition of the events studied here. A sensitivity analysis showed for example that the 1921 event would be pooled with an event occurring in 1922 with a Q80 threshold for the Corrèze river, which is not the case with the

chosen Q90. The choice of the threshold also somewhat conditions the appropriate spatial domain for the spatial matching applied in Sect. 3.2.2. This is further discussed in Sect. 6.2.2.

## 6.2    Spatial matching sensitivity

Section 3.2.2 introduced the method developed to spatially match local events, in the deterministic case. This section aims at summarising the sensitivity of this matching process to different parameters.

### 6.2.1    Sensitivity to the spatial domain

The spatial matching is done considering a specific set of gauging stations as spatial domain. Different spatial domains (for example entire France, or by HER, or by a specific set of stations) would lead to different spatio-temporal event definitions. Consequently, this matching should be preferably done on a consistent set of stations along time to have an homogeneous definition of events. This is the reason why the spatial matching had not been applied to streamflow observations that have an

evolving network – less than 30 stations before the 1910s and more than 600 stations after the 1970s – but also missing data.

### 6.2.2    Sensitivity to the threshold defining local events

The spatial matching procedure tends naturally to aggregate in time consecutive and overlapping local low-flow events from different stations. Moreover, the higher the threshold value used for defining local events, the higher the number of such events (See Sect. 6.1). Consequently, the higher the threshold value in mm or m$^3$/s, the greater the risk to aggregate potentially

independent spatio-temporal events.

### 6.2.3    Consequences on spatial matching limitations

Figure 16 (right) shows that the choices made here – Q90 threshold + two-step (HERs then France) spatial matching – might lead to too much spatio-temporal aggregation for specific events like 1878 and 1976. For these two events, a possible overconfident aggregation might indeed reveals itself through the differentiation of stations in two groups by their start dates.

When looking more into details in the reconstructed 1976 event, it appears that it had two main components in summer 1975 and summer 1976 that were linked together throughout winter 1975/1976 for stations mainly located in the northern half of the country (not shown). This very dry winter in Northern France has been largely documented (Brochet, 1977; Vidal et al., 2010b), and has led to extreme winter and spring low-flows in snow-dominated catchments – visible as pale green filled circles in Fig. 16 (third row, right) – notably in the Alps (Vivian, 1977). This also prevented severely hit northern catchments from

late summer 1975 on to fully recover during the regular recharge season and thus to experience an extreme low-flow event




spanning more than a year (Bremond, 1976; Gazelle, 1977). This analysis also helps interpreting the north-south difference in duration and severity shown in Fig. 16. It thus validates the spatio-temporal aggregation of this event found here with the specific choices of threshold and spatial domain(s).

The 1878 event also has two temporal peaks, but available references and hydrometric data related to this specific event are
much too scarce to finely assess the spatio-temporal aggregation from SCOPE Hydro. Precipitation observation summaries however suggest that the second half-year of 1877 was very dry all around the Mediterranean and especially in the Eastern Pyrenees (Moureaux, 1880a, p. 22). This region remained relatively dry throughout 1878, and especially during February (Moureaux, 1880b, p. 10). The month of September 1878 was then exceptionally dry over a large part of France, including not only the Mediterrean area, but also the Pyrenees and the Massif Central (Mascart, 1880, p. 33), leading to a second component
of the spatio-temporal extreme low-flow event.

Beyond these two specific cases, it has to be noted that both the 1893 and the 1990 events appear quite satisfactorily defined in space and time by the spatial matching procedure proposed here (see Fig. 16, rows 2 and 4.).

## 6.3 Ensemble matching sensitivity

Section 3.2.3 introduced *average events* as a way of summarizing local events from all ensemble members for applying the
spatial matching procedure. Events from individual ensemble members that do not overlap temporally with average events are removed from the final dataset.

In Fig. 7, very short events detected by only a few ensemble members during May 1990 for Station 1 in HER 11 (shown in grey) are for example adequately removed in the final spatio-temporal event definition. However, events detected by 8 ensemble members before July 1989 are also removed – the threshold for deriving an average event being set at 10 for this
station – while being detected in the Safran Hydro dataset. Even if the threshold definition is calibrated against Safran Hydro summary statistics (total duration of events during a given period), it may thus introduce some limited discrepancies between the two datasets. It has to be noted that this particular discrepancy is only local: Fig. 4(e) shows that an event occurred in early 1989 for 12 out of 16 stations in HER 11 in the Safran Hydro dataset, and an event concurrently occurred in 8 stations for average events in the SCOPE Hydro dataset (not shown). The two datasets therefore agree well in the occurrence of an event
during this period over a large part of this HER, even if differences may occur locally due to the local sensitivity of ensemble matching choices.

Fig. 7(b) shows another case of local discrepancy between the two datasets for Station 1 in HER 11: the late 1990 event is clearly detected locally by SCOPE Hydro – by 23 out of 25 ensemble members – but not by Safran Hydro. However, Fig. 4(e) shows that Station 1 is the only one (out of 16) in HER 11 where no event is detected during this period in the Safran Hydro
dataset. Moreover, 15 stations in this HER are hit by an event around this period in the SCOPE Hydro dataset (not shown). This shows once again that the two datasets agree well when considering the regional occurrence of low-flow events.

Figure 7(b) also features a favourable case where the late 1991 event (shown in green) is detected by 23 SCOPE Hydro ensemble members as well as by Safran Hydro. The cases discussed above thus perfectly illustrate the compromise reached here to derive a deterministic series of events from an ensemble of series and to then apply the spatial matching procedure.



## 6.4 Comparison of spatio-temporal events from two different datasets

The previous section briefly touched upon a comparison of extreme low-flow events derived from the Safran Hydro dataset and the SCOPE Hydro dataset, at the local scale, but also in a spatio-temporal view. One may then want to draw a more formal comparison of spatio-temporal events between the two data sets (and possibly with a spatial matching of event derived from observations, but see comments in Sect. 6.2.1). This section discusses further the limitations for automating such comparisons.

The simplest approach would consist in identifying locally previously defined spatio-temporal events through a date overlapping procedure. However, this can lead to identify multiple spatio-temporal events from one dataset to only one event from the second dataset, calling into question the initial spatial matching done (independently) for each dataset. Moreover, this approach would identify events only locally and through a temporal dimension, not considering the spatial extent of the event at the scale of France, which would be too restrictive in practice as shown by the above examples. One response would be to formally identify a spatio-temporal event from one dataset to a spatio-temporal event from another dataset. Some further technical developments would thus have to be set up and tested for comparing any two spatio-temporal extents of extreme low-flow events in both a temporal and spatial dimension.

When some temporal overlapping rules may rather easily be set up at the local scale, corresponding rules for allowance in spatial extent would be much harder to formalise due to the irregular location of hydrometric stations (and upstream catchments). One way forward would be to start the identification based on HER-representative events, which carry less spatio-temporal variability, and report this identification to the local scale. Lastly, one would also need to deal with the ensemble property of the SCOPE Hydro dataset. Again, average events as defined in Sect. 3.2.3 may be the right way forward. Nevertheless, wrongly identifying multiple spatio-temporal events from one dataset to only one event from the second dataset is still a possibility.

## 6.5 Use of median event characteristics

Figures 14, 15 and 16 display median characteristics – duration and severity – in order to show summary results at the scale of France. Nevertheless, Fig. 17 showed a strong variability between events from each individual ensemble member. Taking maximum characteristics instead of median characteristics would lead to very different results in Sect. 5. This is illustrated in Fig. 13 by grey and black points, that represent characteristics from individual ensemble members and median characteristics, respectively. Extreme grey points do not necessarily correspond to stand-out median event characteristics. For example, the grey point plotted for the Ubaye river with a duration of approximately 500 days and a severity of approximately 3.5 mm corresponds to the 1958 spatio-temporal event, which is not labelled as an extreme event based on its median characteristics. Benchmark events shown in Fig. 15 could alternatively be determined based on these maximum characteristics and used for adaptation purpose scenarios. The ensemble characteristics of each spatio-temporal event should therefore be kept as far as possible along the analysis. Indeed, this ensemble reconstruction provides 25 plausible spatio-temporal events corresponding to specific synoptic conditions as given by the 20CR reanalysis. This can be used to better understand the relationship between meteorological conditions and extreme low-flow events.



## 6.6 Uncertainties in reconstructed streamflow

The above section dealt with the uncertainty related to the statistical downscaling step in the SCOPE approach. Several other sources of uncertainty in streamflow reconstructions are not considered in this work. First, all results presented here are conditional on the large-scale information provided by the 20CR atmospheric reanalysis, and using an alternative extended reanalysis like ERA-20C may lead to different outputs. Second, the hydrometeorological modelling chain used here for deriving the SCOPE Hydro dataset – SCOPE and GR6J-Cemaneige – is only one of the many choices for reconstructing high-resolution meteorological fields and catchment-scale streamflow. In order to asses the uncertainty related to this choice, a follow-up study will compare the SCOPE-Hydro dataset (1) to 20CR-driven reconstructions made by Dayon et al. (2015) with a different downscaling method and the physically-based Isba-Modcou hydrological chain, and (2) to reconstructions mixing their downscaling method and the GR6J+Cemaneige hydrological model. Lastly, both above mentioned reconstructions assume a constant land cover and land use, while the wooded area for example is known to have strongly evolved in France since the 19th century (see e.g. Koerner et al., 2000).

## 7 Conclusions

This paper describes an ensemble reconstruction of spatio-temporal extreme low-flow events since 1871 over 662 near-natural catchments in France based on reconstructed climate and streamflow from the SCOPE Climate (Spatially COherent Probabilistic Extended Climate) and SCOPE hydro (Spatially COherent Probabilistic Extended Hydrological) datasets. SCOPE Climate builds on a probabilistic downscaling of the 20CR reanalysis over France as described by Caillouet et al. (2016). This ensemble high-resolution daily meteorological dataset is used as forcings for the GR6J hydrological model – together with the Cemaneige snow-accounting model – to derive the SCOPE Hydro 25-member ensemble daily reconstructed streamflow dataset over a large set of catchments. These two consistent datasets may thus be used for various hydrometeorological studies – and not only for low flows –, by extending the limited historical depth of surface meteorological and streamflow observations currently available in databases (see examples in Fig. 8 and Fig. 9 for streamflow and Caillouet et al. (2016) for precipitation and temperature).

This work proposes an innovative analysis of extreme low-flow events from the SCOPE Hydro based on two distinct features. First, extreme low-flow events are defined locally using the Sequent Peak Algorithm with a combination of a constant threshold and a daily variable threshold. This allows retaining periods with both absolute low-flow values and values significantly lower than the average seasonal cycle, therefore combining the advantages of the two types of thresholds. Second, a spatial matching procedure is developed to identify events both in time and across a set of stations, thus extending previous approaches that could only be applied to data continuous in space like gridded meteorological drought indicators. This spatial matching procedure is then adapted to the ensemble case as provided by SCOPE Hydro.

The above low-flow analysis features are then applied to the SCOPE Hydro dataset to derive and intercompare characteristics of individual spatio-temporal extreme low-flow events at the country scale. For the first time, these events are qualified and compared in a spatially and temporally homogeneous way over 140 years on a large set of catchments. Results bring forward




well-known recent events like 1976 or 1989-1990, but also older and relatively forgotten ones like the 1878 and 1893 events. These results contribute to improve our knowledge on historical events by taking advantage of the higher historical depth of upper-air atmospheric data compared to hydrometric data, and to derive benchmarks events over France. Results also highlight that the worst events in terms of local severity and duration, or spatial extent often belong to the pre-1950 period. This strongly

suggests to reconsider the common use of only very recent events like 2003 as references for building worst-case scenarios for climate change adaptation strategies. Such worst-case scenarios could instead be derived from historical benchmark events, and more specifically from events in any individual ensemble member in the SCOPE Hydro reconstructions, taken as a plausible hydrological consequence of specific large-scale atmospheric situations. They may also help to put recent events and their socio-economic impacts like the 2015 event into a deeper historical perspective, without resorting only to meteorological

drought indicators as proxys for historical low-flows (Van Lanen et al., 2016). This study moreover allows for further detailed analyses of the effect of climate variability and anthropogenic climate change on low-flow hydrology at the scale of France over the last 140 years, by for example extending recent works on the influence of decadal variability on trends (Hannaford et al., 2013).

**Author contribution**

All co-authors collectively designed the experiments. L. Caillouet and A. Devers developed the model code and L. Caillouet performed the simulations. L. Caillouet prepared the manuscript with contributions from all co-authors.

**Appendix A:  Observed and simulated interannual regimes for the two example catchments**

The interannual regime of the two catchment case-studies are presented in Fig. 18. The Corrèze catchment with an oceanic-dominated regime has a summer low-flow period. The Ubaye catchment with a snowmelt-dominated regime has a main low-

flow period in winter and a second one at the end of the summer. The simulated seasonal cycles match generally well the observed ones, with some discrepancies for the Ubaye catchment during the snowmelt period, probably due to remaining upstream reservoir operation management.

**Appendix B:  Synthetic diagram for the SCOPE method**

The SCOPE method summarised by Fig. 19 includes the SANDHY-SUB method developed by Caillouet et al. (2016), the bias

correction step, and the Schaake Shuffle step.

**Appendix C:  Hydro-Ecoregions**

Figure 20 shows the 22 HERs used for in the spatial matching procedure. HER 1 ("Pyrénées" and HER 2 ("Alpes internes") include high mountainous areas whereas HER 3 to HER 5 ("Massif Central sud", "Vosges", "Jura-Préalpes du Nord") in-





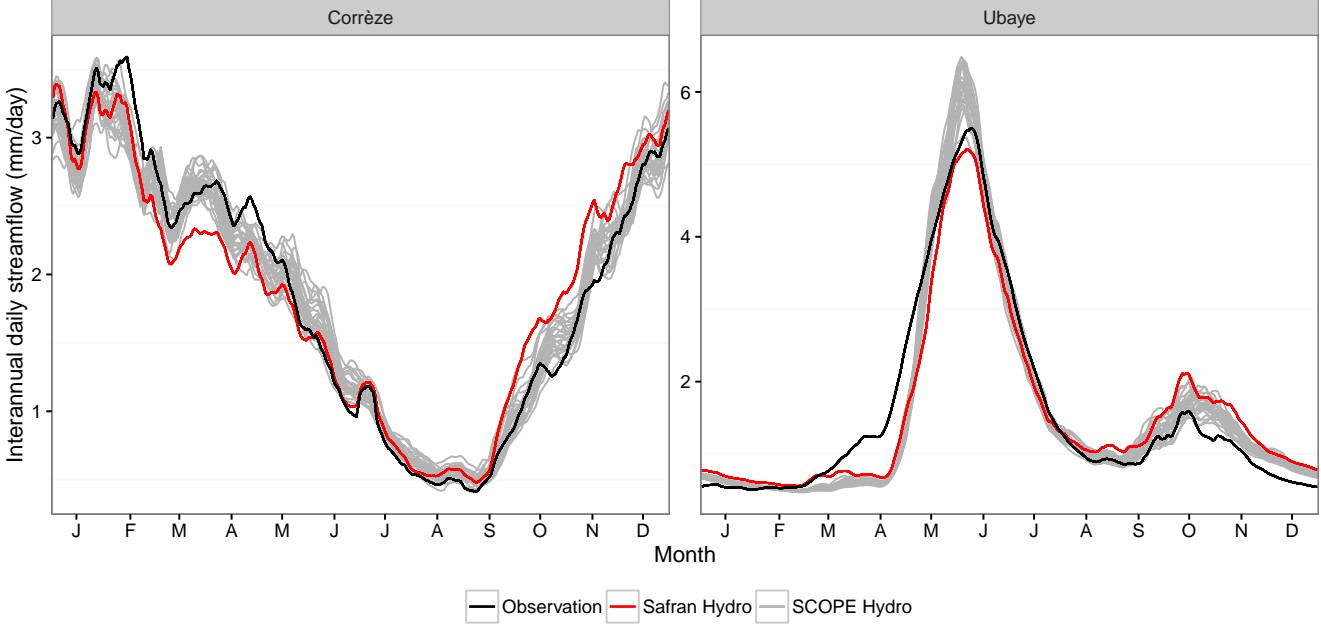

**Figure 18.** Daily interannual average streamflow – from observations, Safran Hydro and SCOPE Hydro – over the CAL period for the Corrèze and Ubaye rivers, smoothed by a 10-day moving average.

clude mountainous areas with lower elevations. HER 6 to HER 8 ("Méditerranéen", "Préalpes du Sud", "Cévennes") include Mediterranean areas where convective precipitation events often occur in Autumn.

## Appendix D: Synthetic diagram for the spatial matching procedure

The overall spatial matching procedure described in Sect. 3.2.2 is synthesised in Fig. 21. A first matching is done within

5   HERs considering an overlap of dates across stations inside each HER. This matching is applied to each HER to derive HER-representative events computed using the median start and end dates of each spatio-temporal event defined within each HER. A second matching is then done considering an overlap of dates across the 22 HER-representative events to identify spatio-temporal events at the scale of France. The inter-HER matching is then reported to the local stations, giving the final France-wide matching.

10   *Acknowledgements.*  The authors would like to thank Météo-France for providing access to the Safran database. The authors would like to thank Guillaume Thirel for his constructive advices provided during the hydrological modelling step. Analyses were performed in R (R Core Team, 2016a), with packages dplyr (Wickham and Francois, 2015), ggplot2 (Wickham, 2009), ggrepel (Slowikowski, 2016), cowplot (Wilke, 2016), RColorBrewer (Neuwirth, 2014), reshape2 (Wickham, 2007), sp (Pebesma and Bivand, 2005; Bivand et al., 2013), grid (R





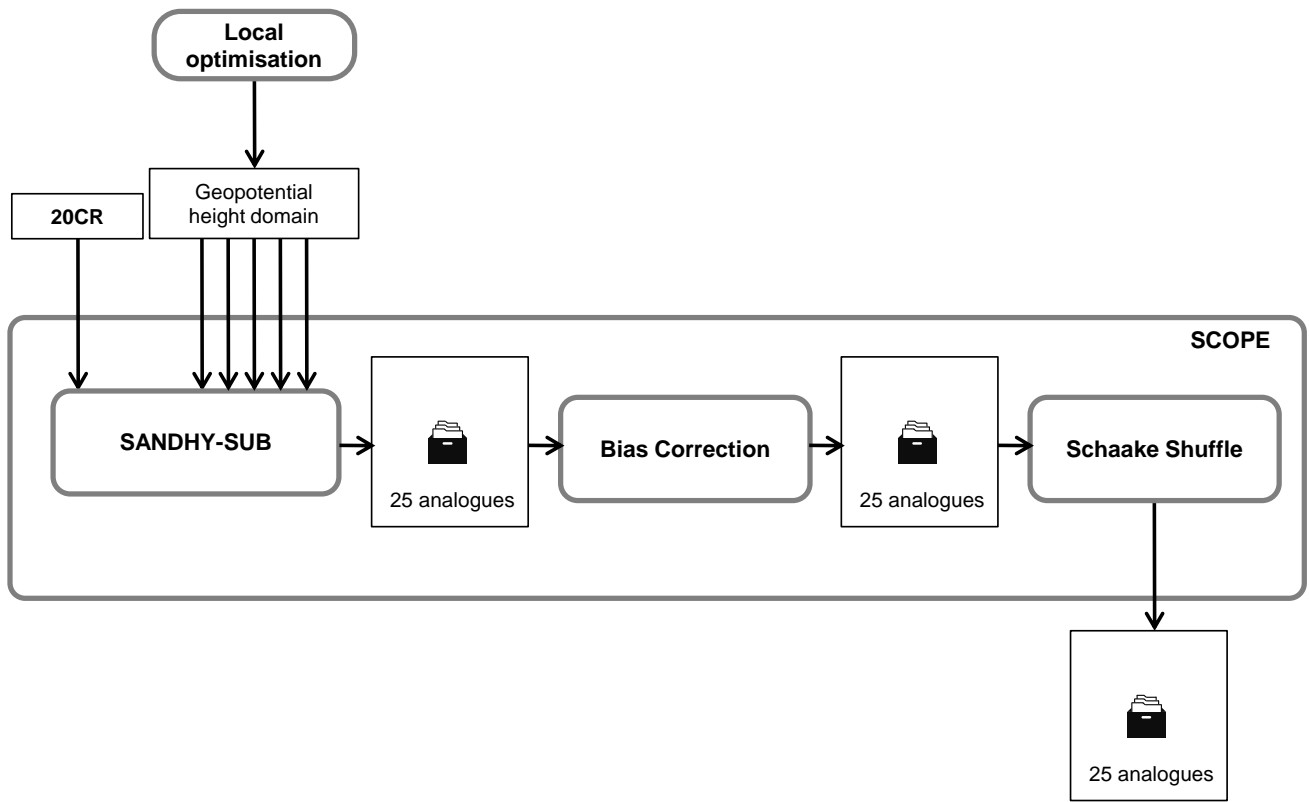

**Figure 19.** Synthetic diagram showing the sequence of steps for the SCOPE method and its use for (20CR-driven) SCOPE Climate reconstructions. See Sect. 2.2.2 for details.

Core Team, 2016b), gridExtra (Auguie, 2016), scales (Wickham, 2016) and lubridate (Grolemund and Wickham, 2011). L. Caillouet PhD thesis is funded by Irstea and CNR.





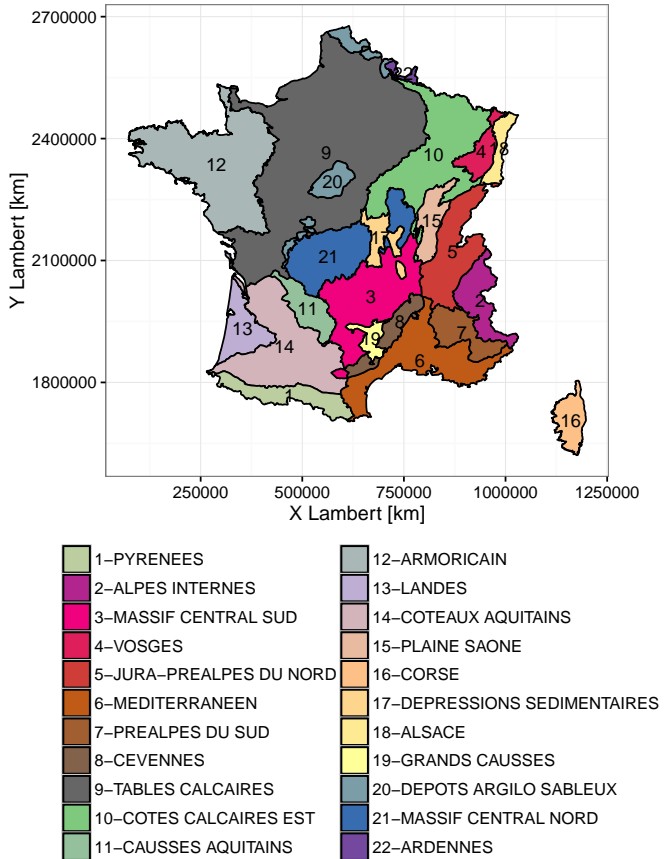

**Figure 20.** Map of the 22 Hydro-ecoregions, adapted from Wasson et al. (2002).

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





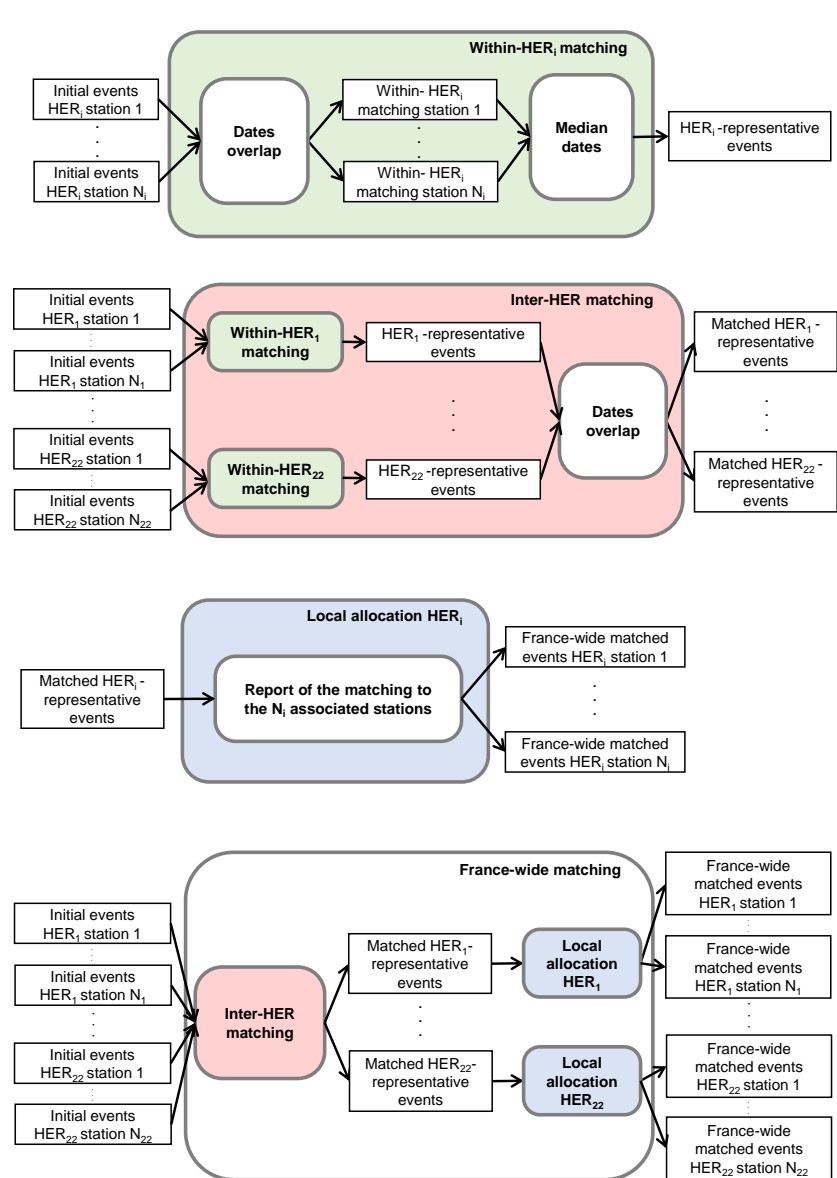

**Figure 21.** Synthetic diagram detailing the spatial matching procedure presented in Sect. 3.2.2.



Ben Daoud, A., Sauquet, E., Bontron, G., Obled, C., and Lang, M.: Daily quantitative precipitation forecasts based on the analogue method: Improvements and application to a French large river basin, Atmos. Res., 169, 147–159, doi:10.1016/j.atmosres.2015.09.015, 2016.

Bivand, R. S., Pebesma, E., and Gomez-Rubio, V.: Applied spatial data analysis with R, Second edition, Springer, NY, http://www.asdar-book.org/, 2013.

Bonaccorso, B., Peres, D. J., Cancelliere, A., and Rossi, G.: Large Scale Probabilistic Drought Characterization Over Europe, Water Resour. Manag., 27, 1675–1692, doi:10.1007/s11269-012-0177-z, 2013.

Bremond, R.: L'incidence du déficit de pluie sur l'écoulement des rivières, Ponts et Chaussées Magazine, 12, 49–58, 1976.

Brigode, P., Brissette, F., Nicault, A., Perreault, L., Kuentz, A., Mathevet, T., and Gailhard, J.: Streamflow variability over 1881-2011 period in northern Quebec: comparison of hydrological reconstructions based on tree rings and on geopotential height field reanalysis, Clim. Past Discuss., doi:10.5194/cp-2016-5, 2016.

Brochet, P.: The 1976 drought in France: climatological aspects and consequences, Hydrological Sciences Bulletin, 22, 393–411, doi:10.1080/02626667709491733, 1977.

Burn, D. H.: Catchment similarity for regional flood frequency analysis using seasonality measures, J. Hydrol., 202, 212–230, doi:10.1016/S0022-1694(97)00068-1, 1997.

Caillouet, L., Vidal, J.-P., Sauquet, E., and Graff, B.: Probabilistic precipitation and temperature downscaling of the Twentieth Century Reanalysis over France, Clim. Past, 12, 635–662, doi:10.5194/cp-12-635-2016, 2016.

Catalogne, C., Sauquet, E., and Lang, M.: Using spot gauging data to estimate the annual minimum monthly flow with a return period of 5 years, Houille Blanche, pp. 78–87, doi:10.1051/lhb/2014042, 2014.

Chauveau, M., Chazot, S., Perrin, C., Bourgin, P.-Y., Sauquet, E., Vidal, J.-P., Rouchy, N., Martin, E., David, J., Norotte, T., Maugis, P., and de Lacaze, X.: What will be impacts of climate change on surface hydrology in France by 2070?, Houille Blanche, pp. 1–15, doi:10.1051/lhb/2013027, 2013.

Chauveau, M., Garcia, J., Mahé, M., and Chazot, S.: Étude de la gestion quantitative et des débits du Rhône en période de « basses eaux », Rapport brli. phase 2. étude des étiages historiques, reconstitution des débits désinfluencés et évaluation de l'empreinte des influences anthropiques sur les débits du rhône. document a – rapport principal de mission 1 : étude des étiages historiques, Agence de l'Eau Rhône-Méditerranée-Corse, 2014.

Cipriani, T., Toilliez, T., and Sauquet, E.: Estimating 10 year return period peak flows and flood durations at ungauged locations in France, Houille Blanche, pp. 5–13, doi:10.1051/lhb/2012024, 2012.

Clark, M., Gangopadhyay, S., Hay, L., Rajagopalan, B., and Wilby, R.: The Schaake Shuffle: A Method for Reconstructing Space-Time Variability in Forecasted Precipitation and Temperature Fields, J. Hydrometeorol., 5, 243–262, doi:10.1175/1525-7541(2004)005<0243:TSSAMF>2.0.CO;2, 2004.

Compo, G. P., Whitaker, J. S., Sardeshmukh, P. D., Matsui, N., Allan, R. J., Yin, X., Gleason, B. E., Vose, R. S., Rutledge, G., Bessemoulin, P., Brönnimann, S., Brunet, M., Crouthamel, R. I., Grant, A. N., Groisman, P. Y., Jones, P. D., Kruk, M. C., Kruger, A. C., Marshall, G. J., Maugeri, M., Mok, H. Y., Nordli, Ø., Ross, T. F., Trigo, R. M., Wang, X. L., Woodruff, S. D., and Worley, S. J.: The Twentieth Century Reanalysis Project, Q. J. Roy. Meteor. Soc., 137, 1–28, doi:10.1002/qj.776, 2011.

Cook, E. R., Seager, R., Kushnir, Y., Briffa, K. R., Büntgen, U., Frank, D., Krusic, P. J., Tegel, W., van der Schrier, G., Andreu-Hayles, L., Baillie, M., Baittinger, C., Bleicher, N., Bonde, N., Brown, D., Carrer, M., Cooper, R., Čufar, K., Dittmar, C., Esper, J., Griggs, C., Gunnarson, B., Günther, B., Gutierrez, E., Haneca, K., Helama, S., Herzig, F., Heussner, K.-U., Hofmann, J., Janda, P., Kontic, R., Köse, N., Kyncl, T., Levanič, T., Linderholm, H., Manning, S., Melvin, T. M., Miles, D., Neuwirth, B., Nicolussi, K., Nola, P., Panayotov,





M., Popa, I., Rothe, A., Seftigen, K., Seim, A., Svarva, H., Svoboda, M., Thun, T., Timonen, M., Touchan, R., Trotsiuk, V., Trouet, V., Walder, F., Ważny, T., Wilson, R., and Zang, C.: Old World megadroughts and pluvials during the Common Era, Science Advances, 1, doi:10.1126/sciadv.1500561, 2015.

Crooks, S. M. and Kay, A. L.: Simulation of river flow in the Thames over 120 years: Evidence of change in rainfall-runoff response?, J. Hydrol., 4, 172–195, doi:10.1016/j.ejrh.2015.05.014, 2015.

Dayon, G.: Evolution du cycle hydrologique sur la France au cours des prochaines décennies, Ph.D. thesis, Université Joseph Fourier, Grenoble, 2015.

Dayon, G., Boé, J., and Martin, E.: Transferability in the future climate of a statistical downscaling method for precipitation in France, J. Geophys. Res-Atmos., 120, 1023–1043, doi:10.1002/2014JD022236, 2015.

Dehérain, P.-P.: La sécheresse en 1893 et la disette de fourrages, Revue des Deux Mondes, 119, 850–881, 1893.

Demargne, J., Wu, L., Regonda, S. K., Brown, J. D., Lee, H., He, M., Seo, D.-J., Hartman, R., Herr, H. D., Fresch, M., Schaake, J., and Zhu, Y.: The science of NOAA's operational hydrologic ensemble forecast service, B. Am. Meteorol. Soc., 95, 79–98, doi:10.1175/BAMS-D-12-00081.1, 2014.

Duband, D.: Rainfall-run-off retrospective of extremes droughts since 1860 in Europe (Germany, Italia, France, Rumania, Spain, Switzerland), Houille Blanche, pp. 51–59, doi:10.1051/lhb/2010041, 2010.

Fleig, A. K., Tallaksen, L. M., Hisdal, H., and Demuth, S.: A global evaluation of streamflow drought characteristics, Hydrol. Earth Syst. Sc., 10, 525–552, doi:10.5194/hess-10-535-2006, 2006.

Forzieri, G., Feyen, L., Rojas, R., Flörke, M., Wimmer, F., and Bianchi, A.: Ensemble projections of future streamflow droughts in Europe, Hydrol. Earth Syst. Sc., 18, 85–108, doi:10.5194/hess-18-85-2014, 2014.

Gazelle, F.: La sécheresse de 1976 en Aquitaine orientale et dans le sud du Massif Central, Revue Géographique des Pyrénées et du Sud-Ouest, 48, 245–268, doi:10.3406/rgpso.1977.3514, 1977.

Giuntoli, I., Renard, B., Vidal, J.-P., and Bard, A.: Low flows in France and their relationship to large-scale climate indices, J. Hydrol., 482, 105–118, doi:10.1016/j.jhydrol.2012.12.038, 2013.

Giuntoli, I., Vidal, J.-P., Prudhomme, C., and Hannah, D. M.: Future hydrological extremes: the uncertainty from multiple global climate and global hydrological models, Earth Syst. Dynam., 6, 267–285, doi:10.5194/esd-6-267-2015, 2015.

Grolemund, G. and Wickham, H.: Dates and Times Made Easy with lubridate, Journal of Statistical Software, 40, 1–25, http://www.jstatsoft.org/v40/i03/, 2011.

Gupta, H. V., Kling, H., Yilmaz, K., and Martinez, G. F.: Decomposition of the mean squared error and NSE performance criteria: Implications for improving hydrological modelling, J. Hydrol., 377, 80–91, doi:10.1016/j.jhydrol.2009.08.003, 2009.

Hannaford, J. and Marsh, T.: An assessment of trends in UK runoff and low flows using a network of undisturbed catchments, Int. J. Climatol., 26, 1237–1253, doi:10.1002/joc.1303, 2006.

Hannaford, J., Buys, G., Stahl, K., and Tallaksen, L. M.: The influence of decadal-scale variability on trends in long European streamflow records, Hydrol. Earth Syst. Sc., 17, 2717–2733, doi:10.5194/hess-17-2717-2013, 2013.

Hisdal, H., Stahl, K., Tallaksen, L. M., and Demuth, S.: Have streamflow droughts in Europe become more severe or frequent?, Int. J. Climatol., 21, 317–333, doi:10.1002/joc.619, 2001.

Jones, P. D., Lister, D. H., Wilby, R. L., and Kostopoulou, E.: Extended riverflow reconstructions for England and Wales, 1865-2002, Int. J. Climatol., 26, 219–231, doi:10.1002/joc.1252, 2006.





Koerner, W., Cinotti, B., Jussy, J.-H., and Benoît, M.: Évolution des surfaces boisées en France depuis le début du XIXᵉ siècle : identification et localisation des boisements des territoires agricoles abandonnés, Revue Forestière Française, 52, 249–269, doi:10.4267/2042/5359, 2000.

Kuentz, A., Mathevet, T., Gailhard, J., Perret, C., and Andréassian, V.: Over 100 years of climatic and hydrologic variability of a mediterranean and mountainous watershed: The Durance River, in: Cold and mountain region hydrological systems under climate change: towards improved projections, edited by Gelfan, A., Yang, D., Gusev, Y., and Kunstmann, H., vol. 360 of *IAHS Red Books*, pp. 19–25, IAHS, 2013.

Kuentz, A., Mathevet, T., Gailhard, J., and Hingray, B.: Building long-term and high spatio-temporal resolution precipitation and air temperature reanalyses by mixing local observations and global atmospheric reanalyses: the ANATEM method, Hydrol. Earth Syst. Sc., 19, 2717–2736, doi:10.5194/hess-19-2717-2015, 2015.

Laaha, G., Gauster, T., Tallaksen, L. M., Vidal, J.-P., Stahl, K., Prudhomme, C., Heudorfer, B., Vlnas, R., Ionita, M., Van Lanen, H. A. J., Adler, M.-J., Caillouet, L., Delus, C., Fendekova, M., Gailliez, S., Hannaford, J., Kingston, D., Van Loon, A. F., Mediero, L., Osuch, M., Romanowicz, R., Sauquet, E., Stagge, J. H., and Wong, W. K.: The European 2015 drought from a hydrological perspective, Hydrol. Earth Syst. Sci. Discussions, doi:10.5194/hess-2016-366, 2016.

Labuhn, I., Daux, V., Girardclos, O., Stievenard, M., Pierre, M., and Masson-Delmotte, V.: French summer droughts since 1326 AD: a reconstruction based on tree ring cellulose $\delta^{18}$O, Clim. Past, 12, 1101–1117, doi:10.5194/cp-12-1101-2016, 2016.

Le Gros, C., Sauquet, E., Lang, M., Achard, A.-L., Leblois, E., and Biton, B.: The hydrological yearbooks published by the Société Hydrotechnique de France: a valuable source of information on hydrology in France, Houille Blanche, pp. 66–77, doi:10.1051/lhb/20150048, 2015.

Le Moine, N.: Le bassin versant de surface vu par le souterrain : une voie d'amélioration des performances et du réalisme des modèles pluie-débit ?, Ph.D. thesis, Université Pierre et Marie Curie, 2008.

Lennard, A. T., Macdonald, N., Clark, S., and Hooke, J. M.: The application of a drought reconstruction in water resource management, Hydrol. Res., 47, 1–14, doi:10.2166/nh.2015.090, 2015.

Mascart, E., ed.: Annales du bureau central météorologique de France – Année 1878 – Tome II : Bulletin des observations françaises et revue climatologique, pp. 25–36 (Planches), Gauthiers-Villars, 1880.

McKee, T. B., Doesken, N. J., and Kleist, J.: The relationship of drought frequency andduration to time scales., in: 8th Conference on AppliedClimatology, Am. Meteorol. Soc., Anaheim, Calif, 1993.

Meko, D. M., Woodhouse, C. A., and Morino, K.: Dendrochronology and links to streamflow, J. Hydrol., 412-413, 200–209, doi:10.1016/j.jhydrol.2010.11.041, 2012.

Mérillon, Y. and Chaperon, P.: Drought in France in 1989, Houille Blanche, pp. 325–340, doi:10.1051/lhb/1990025, 1990.

Moreau, F.: Drought crisis management: Loire basin example, Houille Blanche, 4, 70–76, doi:10.1051/lhb:200404010, 2004.

Moureaux, T.: Sur le régime des pluies en France pendant l'année 1877, in: Annales du bureau central météorologique de France – Année 1877 – Pluies en France – Observations publiées avec la coopération du ministère des travaux publics et le concours de l'association scientifique, edited by Mascart, E., Gauthiers-Villars, 1880a.

Moureaux, T.: Sur le régime des pluies en France pendant l'année 1878, in: Annales du bureau central météorologique de France – Année 1878 – Tome III – Pluies en France – Observations publiées avec la coopération du ministère des travaux publics et le concours de l'association scientifique, edited by Mascart, E., Gauthiers-Villars, 1880b.

Neuwirth, E.: RColorBrewer: ColorBrewer Palettes, https://CRAN.R-project.org/package=RColorBrewer, r package version 1.1-2, 2014.





Nicault, A., Boucher, E., Bégin, C., Guiot, J., Marion, J., Perreault, L., Roy, R., Savard, M. M., and Bégin, Y.: Hydrological reconstruction from tree-ring multi-proxies over the last two centuries at the Caniapiscau Reservoir, northern Québec, Canada, J. Hydrol., 513, 435–445, doi:10.1016/j.jhydrol.2014.03.054, 2014.

Nicolle, P., Pushpalatha, R., Perrin, C., François, D., Thiéry, D., Mathevet, T., Le Lay, M., Besson, F., Soubeyroux, J.-M., Viel, C., Regimbeau, F., Andréassian, V., Maugis, P., Augeard, B., and Morice, E.: Benchmarking hydrological models for low-flow simulation and forecasting on French catchments, Hydrol. Earth Syst. Sc., 18, 2829–2857, doi:10.5194/hess-18-2829-2014, 2014.

Pebesma, E. J. and Bivand, R. S.: Classes and methods for spatial data in R, R News, 5, 9–13, http://CRAN.R-project.org/doc/Rnews/, 2005.

Perrin, C., Michel, C., and Andréassian, V.: Improvement of a parsimonious model for streamflow simulation, J. Hydrol., 279, 275–289, doi:10.1016/S0022-1694(03)00225-7, 2003.

Pfister, C., Weingarter, R., and Luterbacher, J.: Hydrological winter droughts over the last 450 years in the Upper Rhine basin : a methodological approach, Hydrolog. Sci. J., 51, 966–985, doi:10.1623/hysj.51.5.966, 2006.

Plumandon, J.-R.: La sécheresse du printemps 1893, La Nature, pp. 27–28, 1893.

Poli, P., Hersbach, H., Dee, D. P., Berrisford, P., Simmons, A. J., Vitart, F., Laloyaux, P., Tan, D. G. H., Peubey, C., Thépaut, J.-N., Trémolet, Y., Hólm, E. V., Bonavita, M., Isaksen, L., and Fisher, M.: ERA-20C: An atmospheric reanalysis of the twentieth century, J. Climate, 29, 4083–4097, doi:10.1175/JCLI-D-15-0556.1, 2016.

Prudhomme, C., Giuntoli, I., Robinson, E. L., Clark, D. B., Arnell, N. W., Dankers, R., Fekete, B. M., Franssen, W., Gerten, D., Gosling, S. N., Hagemann, S., Hannah, D. M., Kim, H., Masaki, Y., Satoh, Y., Stacke, T., Wada, Y., and Wisser, D.: Hydrological droughts in the 21st century, hotspots and uncertainties from a global multimodel ensemble experiment, Proc. Natl. Acad. Sci. U. S. A., 111, 3262–3267, doi:10.1073/pnas.1222473110, 2014.

Pushpalatha, R., Perrin, C., Le Moine, N., Mathevet, T., and Andréassian, V.: A downward structural sensitivity analysis of hydrological models to improve low-flow simulation, J. Hydrol., 411, 66–76, doi:10.1016/j.jhydrol.2011.09.034, 2011.

Pushpalatha, R., Perrin, C., Le Moine, N., and Andréassian, V.: A review of efficiency criteria suitable for evaluating low-flow simulations, J. Hydrol., 420-421, 171–182, doi:10.1016/j.jhydrol.2011.11.055, 2012.

Quintana-Seguí, P., Le Moigne, P., Durand, Y., Martin, E., Habets, F., Baillon, M., Canellas, C., Franchistéguy, L., and Morel, S.: Analysis of near-surface atmospheric variables: validation of the safran analysis over france, J. Appl. Meteorol. Clim., 47, 92–107, doi:10.1175/2007JAMC1636.1, 2008.

R Core Team: R: A Language and Environment for Statistical Computing, R Foundation for Statistical Computing, Vienna, Austria, https://www.R-project.org/, 2016a.

R Core Team: R: A Language and Environment for Statistical Computing, R Foundation for Statistical Computing, Vienna, Austria, https://www.R-project.org/, 2016b.

Radanovics, S., Vidal, J.-P., Sauquet, E., Ben Daoud, A., and Bontron, G.: Optimising predictor domains for spatially coherent precipitation downscaling, Hydrol. Earth Syst. Sc., 17, 4189–4208, doi:10.5194/hess-17-4189-2013, 2013.

Robertson, D. E., Shrestha, D. L., and Wang, Q. J.: Post-processing rainfall forecasts from numerical weather prediction models for short-term streamflow forecasting, Hydrol. Earth Syst. Sc., 17, 3587–3603, doi:10.5194/hess-17-3587-2013, 2013.

Sauquet, E.: Mapping mean annual river discharges: geostatistical developments for incorporating river network dependencies, J. Hydrol., 331, 300–314, doi:10.1016/j.jhydrol.2006.05.018, 2006.

Sauquet, E. and Catalogne, C.: Comparison of catchment grouping methods for flow duration curve estimation at ungauged sites in France, Hydrol. Earth Syst. Sc., 15, 2421–2435, doi:10.5194/hess-15-2421-2011, 2011.



Shiau, J. T. and Shen, H. W.: Recurrence analysis of hydrologic droughts of differing severity, J. Water Res. Pl. Man.-ASCE, 127, 30–40, doi:10.1061/(ASCE)0733-9496, 2001.

Sippel, S., Otto, F. E. L., Forkel, M., Allen, M. R., Guillod, B. P., Heimann, M., Reichstein, M., Seneviratne, S. I., Thonicke, K., and Mahecha, M. D.: A novel bias correction methodology for climate impact simulations, Earth System Dynamics, 7, 71–88, doi:10.5194/esd-7-71-2016, 2016.

Slowikowski, K.: ggrepel: Repulsive Text and Label Geoms for 'ggplot2', https://CRAN.R-project.org/package=ggrepel, r package version 0.5, 2016.

Smakhtin, V. U.: Low flow hydrology: a review, J. Hydrol., 240, 147–186, doi:10.1016/S0022-1694(00)00340-1, 2001.

Snelder, T. H., Datry, T., Lamouroux, N., Larned, S. T., Sauquet, E., Pella, H., and Catalogne, C.: Regionalization of patterns of flow intermittence from gauging station records, Hydrol. Earth Syst. Sc., 17, 2685–2699, doi:10.5194/hess-17-2685-2013, 2013.

Soubeyroux, J.-M., Vidal, J.-P., Baillon, M., Blanchard, M., Céron, J.-P., Franchistéguy, L., Régimbeau, F., Martin, E., and Vincendon, J.-C.: Characterizing and forecasting droughts and low-flows in France with the Safran-Isba-Modcou hydrometeorological suite, Houille Blanche, pp. 30–39, doi:10.1051/lhb/2010051, 2010.

Spraggs, G., Peaver, L., Jones, P., and Ede, P.: Re-construction of historic drought in the Anglian Region (UK) over the period 1798–2010 and the implications for water resources and drought management, J. Hydrol., 526, 231–252, doi:10.1016/j.jhydrol.2015.01.015, 2015.

Tallaksen, L. and Stahl, K.: Spatial and temporal patterns of large-scale droughts in Europe: Model dispersion and performance, Geophys. Res. Lett., 41, 429–434, doi:10.1002/2013GL058573, 2014.

Tallaksen, L., Madsen, H., and Clausen, B.: On the definition and modelling of streamflow drought duration and deficit volume, Hydrolog. Sci. J., 42, 15–33, doi:10.1080/02626667709492003, 1997.

Tallaksen, L. M. and Van Lanen, H. A. J.: Hydrological Drought: Processes and estimation methods for streamflow and groundwater, vol. 48 of *Developments in Water Science*, Elsevier, 2004.

Timbal, B., Arblaster, J., and Power, S.: Attribution of the late 20th century rainfall decline in South-West Australia, J. Climate, 19, 2046–2062, doi:10.1175/JCLI3817.1, 2006.

Uhlemann, S., Thieken, A. H., and Merz, B.: A consistent set of trans-basin floods in Germany between 1952–2002, Hydrol. Earth Syst. Sc., 14, 1277–1295, doi:10.5194/hess-14-1277-2010, 2010.

Valery, A.: Modélisation précipitations - débit sous influence nivale. Élaboration d'un module neige et évaluation sur 380 bassins versants., Ph.D. thesis, AgroParisTech, 2010.

Valéry, A., Andréassian, V., and Perrin, C.: 'As simple as possible but not simple': What is useful in a temperature-based snow-accounting routine? Part 2 - Sensitivity analysis of the Cemaneige snow accounting routine on 380 catchments, J. Hydrol., 517, 1176 – 1187, doi:10.1016/j.jhydrol.2014.04.058, 2014.

van Huijgevoort, M. H. J., Hazenberg, P., van Lanen, H. A. J., and Uijlenhoet, R.: A generic method for hydrological drought identification across different climate regions, Hydrol. Earth Syst. Sc., 16, 2437–2451, doi:10.5194/hess-16-2437-2012, 2012.

Van Lanen, H., Laaha, G., Kingston, D. G., Gauster, T., Ionita, M., Vidal, J.-P., Vlnas, R., Tallaksen, L. M., Stahl, K., Hannaford, J., Delus, C., Fendekova, M., Mediero, L., Prudhomme, C., Rets, E., Romanowicz, R. J., Gailliez, S., Wong, W. K., Adler, M.-J., Blauhut, V., Caillouet, L., Chelcea, S., Frolova, N., Gudmundsson, L., Hanel, M., Haslinger, K., Kireeva, M., Osuch, M., Sauquet, E., Stagge, J. H., and Van Loon, A. F.: Hydrology needed to manage droughts: the 2015 European case, Hydrol. Process., doi:10.1002/hyp.10838, 2016.

Van Loon, A. F. and Laaha, G.: Hydrological drought severity explained by climate and catchment characteristics, J. Hydrol., 526, 3–14, doi:10.1016/j.jhydrol.2014.10.059, 2015.





Van Loon, A. F. and Van Lanen, H. A. J.: A process-based typology of hydrological drought, Hydrol. Earth Syst. Sc., 16, 1915–1946, doi:10.5194/hess-16-1915-2012, 2012.

Verkade, J. S., Brown, J. D., Reggiani, P., and Weerts, A. H.: Post-processing ECMWF precipitation and temperature ensemble reforecasts for operational hydrologic forecasting at various spatial scales, J. Hydrol., 501, 73–91, doi:10.1016/j.jhydrol.2013.07.039, 2013.

Vidal, J.-P., Martin, E., Franchistéguy, L., Baillon, M., and Soubeyroux, J.-M.: A 50-year high-resolution atmospheric reanalysis over France with the Safran system, Int. J. Clim., 30, 1627–1644, doi:10.1002/joc.2003, 2010a.

Vidal, J.-P., Martin, E., Franchistéguy, L., Habets, F., Soubeyroux, J.-M., Blanchard, M., and Baillon, M.: Multilevel and multiscale drought reanalysis over France with the Safran-Isba-Modcou hydrometeorological suite, Hydrol. Earth Syst. Sc., 14, 459–478, doi:10.5194/hess-14-459-2010, 2010b.

Vidal, J.-P., Martin, E., Kitova, N., Najac, J., and Soubeyroux, J.-M.: Evolution of spatio-temporal drought characteristics: validation, projections and effect of adaptation scenarios, Hydrol. Earth Syst. Sc., 16, 2935–2955, doi:10.5194/hess-16-2935-2012, 2012.

Vidal, J.-P., Hingray, B., Magand, C., Sauquet, E., and Ducharne, A.: Hierarchy of climate and hydrological uncertainties in transient low flow projections, Hydrol. Earth Syst. Sc. Discussions, 12, 12 649–12 701, doi:10.5194/hessd-12-12649-2015, 2015.

Vimont, E.: La sécheresse en 1893, L'Astronomie, 12, 309–311, 1893.

Vivian, H.: L'hydrologie nord-alpine et la sécheresse de 1976, Revue de Géographie de Lyon, 52, 117–151, doi:10.3406/geoca.1977.1199, 1977.

Vogel, R. M. and Stedinger, J. R.: Generalized Storage-Reliability-Yield Relationships, J. Hydrol., 89, 303–327, doi:10.1016/0022-1694(87)90184-3, 1987.

Vrac, M. and Friederichs, P.: Multivariate – Intervariable, Spatial, and Temporal – Bias Correction, J. Climate, 28, 218–237,
doi:10.1175/JCLI-D-14-00059.1, 2015.

Šípek, V. and Daňhelka, J.: Modification of input datasets for the Ensemble Streamflow Prediction based on large-scale climatic indices and weather generator, J. Hydrol., 528, 720–733, doi:10.1016/j.jhydrol.2015.07.008, 2015.

Wasson, J. G., Chandesris, A., Pella, H., and Blanc, L.: Typology and reference conditions for surface water bodies in France: the hydro-ecoregion approach, in: Typology and ecological classification of lakes and rivers, edited by Ruoppa, M. and Karttunen, K., no. 566 in
TemaNord, pp. 37–41, Nordic Council of Ministers, Copenhagen, Denmark, 2002.

Wickham, H.: Reshaping Data with the reshape Package, Journal of Statistical Software, 21, 1–20, http://www.jstatsoft.org/v21/i12/, 2007.

Wickham, H.: ggplot2: Elegant Graphics for Data Analysis, Springer-Verlag New York, http://ggplot2.org, 2009.

Wickham, H.: scales: Scale Functions for Visualization, https://CRAN.R-project.org/package=scales, r package version 0.4.0, 2016.

Wickham, H. and Francois, R.: dplyr: A Grammar of Data Manipulation, https://CRAN.R-project.org/package=dplyr, r package version
30  0.4.3, 2015.

Wilke, C. O.: cowplot: Streamlined Plot Theme and Plot Annotations for 'ggplot2', https://CRAN.R-project.org/package=cowplot, r package version 0.6.2, 2016.

Wilks, D. S.: Multivariate ensemble Model Output Statistics using empirical copulas, Q. J. Roy. Meteor. Soc., 141, 945–952, doi:10.1002/qj.2414, 2014.

Zaidman, M. D., Rees, H. G., and Young, A. R.: Spatio-temporal development of streamflow droughts in north-west Europe, Hydrol. Earth Syst. Sc., 6, 733–751, doi:10.5194/hess-6-733-2002, 2002.