# Peer review of "Ensemble reconstruction of spatio-temporal extreme low-flow events in France since 1871"

_Hydrology and Earth System Sciences, 2016_

## Referee Comment (RC1) · Anonymous Referee #1 · 13 Oct 2016

**Ensemble reconstruction of spatio-temporal extreme low-flow events in France since 1871 by Caillouet et al.**

In this paper, the authors set out to reconstruct low-flow events in a reference network of catchments across France from 1871 to present. Due to the lack of observed streamflow records prior to the 1950s, downscaled climatological data (20CR) is used to reconstruct streamflow using hydrological models. Using knowledge of more recent low flow events over the past 50 years, the authors validate their novel approaches of low-flow event identification, spatial matching, and hydrological modelling, which succeed in identifying the well-known events of 1976 and 1989-1990. The approach highlights two additional extreme low-flow events in the 19th century: 1878 and 1893 and the authors conclude that many severe, long and widespread drought events occur prior to 1950.

This paper provides a valuable contribution to science in both methodology and results. The methods are many and varied and though multiple subjective decisions are included, they are well thought out and discussed. Whilst the many steps of the spatial matching procedure are quite challenging to decipher, the results are well presented. The combined threshold level is a particularly interesting concept. This is a very long paper, which might daunt the average reader, but the content is valuable and in my opinion the many maps and graphs all convey interesting information. The maps in Figures 15 and 16 are of particular merit, though Figure 15 would benefit from some slight adjustments (see comment below).

If the editor deems the paper too long (which would be understandable), there are several ways in which the main body of this manuscript could be cut down:

1. The introduction could be reduced.
2. The description of the derivation of the SCOPE climate data from 20CR-SANDHY-SUB could be stated much more succinctly in the main text, and a detailed description provided in the appendix.
3. Similarly the description of the spatial matching procedure is lengthy, and could be summarised with a longer description in an appendix. Those who wish to reproduce your methods would be willing to read the appendices, whereas others, who are mainly interested in the results would not need to understand that level of detail.
4. The day of the year analysis in figure 12 could be cut out, however it does show the spread of the start dates.
5. Furthermore, Figure 13 could be removed, as the majority of the information is given in figures 10 and 11. Again however, it does show the linear relationship between severity and duration of events.

**HESS REVIEW CHECKLIST**

1. Does the paper address relevant scientific questions within the scope of HESS?
   *Yes*
2. Does the paper present novel concepts, ideas, tools, or data?
   *Yes*
3. Are substantial conclusions reached?
   *Yes*
4. Are the scientific methods and assumptions valid and clearly outlined?

*SCOPE Climate production and spatial matching require more clarity, see further comments below*

5. Are the results sufficient to support the interpretations and conclusions?
   *Yes*
6. Is the description of experiments and calculations sufficiently complete and precise to allow their reproduction by fellow scientists (traceability of results)?
   *Not quite – See comment 4*
7. Do the authors give proper credit to related work and clearly indicate their own new/original contribution?
   *Yes*
8. Does the title clearly reflect the contents of the paper?
   *Yes*
9. Does the abstract provide a concise and complete summary?
   *Yes*
10. Is the overall presentation well-structured and clear?
    *Mostly, see further comments below*
11. Is the language fluent and precise?
    *Mostly, see further comments below*
12. Are mathematical formulae, symbols, abbreviations, and units correctly defined and used?
    *Except one error in KGE, see comments below*
13. Should any parts of the paper (text, formulae, figures, tables) be clarified, reduced, combined, or eliminated?
    *Yes, see comments below*
14. Are the number and quality of references appropriate?
    *Yes*
15. Is the amount and quality of supplementary material appropriate?
    *Appendices clarifying spatial matching and SCOPE Climate production needed*

**I recommend this paper to be published, subject to minor corrections, as below:**

**Comments that must be addressed:**

1. The derivation of the SCOPE climate data is not well explained. Page 6 needs a lot of attention. It appears to be a significant amount of work that is not published elsewhere. This procedure needs to be clearly described, and moved to the appendix. For example, it is not clear what variables from the 20CR are downscaled (500hPa geopotential height?)
2. Please indicate the ideal value of KGE (1 I assume?) on both the description of the metric on page 7 and the map given as Figure 2. Furthermore, please check the equation provided on page 7, as this indicates that a low value should be a good score. Gupta (2009) provide the equation you present as the Euclidian Distance (*ED*), whilst KGE is 1-*ED*.
3. Your description of the spatial matching procedure needs some work to improve its clarity. I suggest it is lengthened and sent to an appendix.
4. Page 25 Figure 17 – add a scale/legend to this figure.
5. Page 31 lines 3 and 4 – You state "Result also highlight that the worst events in terms of severity and duration or spatial extent often belong to the pre-1950 period". This statement is not backed up by your graphics. Figure 14 indicates the majority of the most widespread events were post 1940, Figure 15 indicates that 1990 and 1976 were the longest, and similarly in figure 13 only 1945 is picked out among the most severe in the Corrèze catchment. Furthermore, figures 10 and 11 do indicate some long and severe droughts prior to 1950, but the latter period on these graphs seem to show significantly more events and

more serious events than the earlier period. I think what you mean to say is that many severe, long and widespread events occur prior to the 1950s, but I don't think it's true that the majority of the most severe events are in the early part of your reconstructions. Please amend this sentence accordingly.

**Further recommended minor adjustments and comments:**

1. Page 2 lines 2 to 6 – references here may be assumed to be based in French catchments due to the outlining of the lack of available data in France, then using the word consequently to start the next sentence. Rephrase to indicate these are not all French studies.
2. Page 3 lines 3 and 4 – "high spatial and temporal scales" and "large-scale atmospheric and oceanic data" high scale means different things to different people. Do you mean high resolution or not?
3. Page 3 line 13 – within which country was Dayon's study based?
4. Page 3 line 16 – downscaling of 20CR – what variables did you downscale and at what resolution?
5. Page 3 line 34 to page 4 line 1 – you mention the use of probabilistic data to account for uncertainty, but very few of your figures display the uncertainty from the ensemble in the results (only really Figures 10 and 11). It would be nice to see some spatial mapping of uncertainty – perhaps a map of one of your events, with density of colour according to how many of the 25 ensemble members identify that event, this map could accompany figure 17.
6. Page 4 line 19 – what does "without direct human influence" mean?!
7. Page 4 line 27 – we are referred to Annex A which mentions Safran Hydro and SCOPE Hydro before the datasets are introduced. I suggest you add "see section 2.2.1 and 2.2.2" to the caption of Figure 18.
8. Page 7 line 20 – I don't think it can be said to give "equal weights", better to say "reduces the bias towards high flows"
9. Page 7 line 24 – how many stations used CemaNeige then? State this. Could you highlight them somehow on Figure 2?
10. Page 7 line 26 – so all the catchments were given the same values for the 2 CemaNeige parameters? Is this realistic? Or were the catchments calibrated several times, and then the median of those given to each individual catchment? This needs to be clearer.
11. Page 8 lines 17-24 – I appreciate that this algorithm comes from reservoir design, but it would be more appropriate to explain it in the context of flow deficit here.
12. Page 8 line 25 - Your *end date* is defined as the time of maximum depletion. Do you think this is representative of the end of an extreme low flow/drought period? What about the time it takes for the stream to return to "normal" flows? See Parry et al (2016) http://ppg.sagepub.com/content/early/2016/06/02/0309133316652801.refs
13. Page 11 Figure 4 – Station 11 shows a single black bar in (a), but a red and a grey bar in (b), is this actually one event or is fig (a) not quite clear enough?
14. Page 14 Figure 7 – why is the late 1990 event not picked up by Safran?
15. Page 14 Figure 7 – this figure takes a very long time to load on my PC – it interrupts scrolling significantly, and almost crashes my browser. Is its file size much larger than Figure 4? If so why? Can it be reduced without compromising its quality?
16. Captions for Figures 8 and 9 – don't think you need to point out the different scales on the y-axes.
17. Page 16 – provide an assessment of the bias in SCOPE Hydro (median) compared with observed flows, for low and high flow seasons. Calculate the percentage of time the obs are within the SCOPE Hydro range for all stations.

18. Page 17 line 12 – not sure I agree that the observations are "most of the time" included in the SCOPE Hydro range – the obs seem to be on the periphery most of the time – address with comment above.
19. Page 18 Figure 10 and page 20 Figure 12 – the dashed lines are visible on the computer, but not on the print out for me. This could easily be a problem my end, but please double check it prints correctly for you.
20. Page 18 figure 10 – the "Whiskers" (note spelling!) do not extend to 1.5 times the IQR, they extend to the largest (and smallest) observations still within 1.5 times the IQR
21. Page 19 line 17 – "A higher number of extreme events and higher severity values are simulated after 1940 for the Corrèze" – do you have any stats to prove this? Mann-Whitney U test for step change? Is there statistical significance?
22. Page 20 line 9 – "There is no visible trend on the seasonality of start dates" – did you do any statistical tests?
23. Page 22 Figure 14 – You have displayed months and years on this plot whilst all other plots just categorise by year. You could remove the months from this plot to make the plot clearer.
24. Page 23 Figure 15 – this is a great Figure, but it really isn't colour-blind friendly. Remember 1 in 10 men is colour-blind. I know this will be difficult to take into account with a map like this, but I suggest you make sure that the 3 or 4 main events you are picking out are in contrasting colours that colour-blind people can differentiate from the others. At the moment, for those with Protanope colour-blindness (the most common) 1990 is clearly visible, 1893 and 1985 look the same as each other, 1878 and 1978 look the same as each other, 1943 and 1949 look the same as each other, and the rest are all very similar. Try using the mobile app "CVSimulator" to test your images (I've run your image through this app, see graphic at end of document). On this note – Figures and 4 and 7 use orange and green which are indistinguishable to the colour-blind. Reconsider this colouring if possible. Figures 16 and 17 are OK, Figure 20 isn't but I don't think the issue can be avoided here sadly.
25. Page 23 line 4 – I think you have 1878 and 1893 the wrong way round in this sentence
26. Page 23 lines 7 to 9 – comment on the 2003 event – this is a major finding, I suggest you add this to the conclusion!
27. Page 26 lines 4 to 9 – this discussion seemed to suggest that 1990 wasn't a particularly severe event, whereas Figure 16 indicates it really was just as severe as the 1893 and 1976 events, especially in the massif central regions.
28. Page 26 lines 10 and 11 – you state that the 1990 event start dates lie within a common 6 month period for all stations concerned – however I can definitely see some green dots on that lower right map in Figure 16! I suggest rewording "all stations concerned" as "for the vast majority of stations"
29. Page 30 section 6.6 – no mention of parameter uncertainty and model structural uncertainty – e.g. GR4J and GR5J.
30. Page 30 conclusion – no mention of the use of Safran Hydro

**Spelling/Grammatical Errors:**

1. Page 1 line 1 – replace "historical depth" with "period" or "length"
2. Page 1 line 7 – change "downscaling of the" to "downscaled from the"
3. Page 1 line 16 – replace first word "on" with "of"
4. Page 1 line 25 – change "a deep knowledge" to "a comprehensive knowledge"
5. Page 2 line 4 – remove the word "only"

6. Page 2 line 6 – replace "or on few stations available on longer periods" to "or on a few stations available for longer periods"
7. Page 2 line 8 – replace "breadth and depth" with "spatial coverage and record lengths"
8. Page 2 line 13 – I would use the word "recovered" rather than "reconstituted" in this context
9. Page 2 line 19 – replace "it has to be noted" with "it should be noted"
10. Page 3 line 9 – replace "reconstructions on the 20$^{th}$" with "reconstructions of the 20$^{th}$"
11. Page 3 line 11 – replace "depth" with "extent"
12. Page 3 line 11 – replace "based on a analogue" with "based on an analogue" – check remaining script for correct usage of "a" and "an"
13. Page 3 line 14 – replace "combined to a physically" with "combined with a physically"
14. Page 4 line 21 – add an "s" to influences and remove "a" for "with good quality"
15. Page 5 line 14 – replace "paving" with "covering"
16. Page 6 line 35 – replace "19 diagram" with "Figure 19"
17. Page 8 line 18 – replace "if" with "is"
18. Page 10 line 17 – "would lead to too much aggregated events", do you mean "would lead to too many aggregated events"? If not reword for clarity.
19. Page 10 lines 27 to 29 – this sentence doesn't make sense – it needs re-wording.
20. Page 10 line 30 – replace "earlier" with "earliest"
21. Throughout manuscript – "Cemaneige" written in manuscript, "CemaNeige" is naming convention in IRSTEA documentation.
22. Page 12 lines 19 and 20 – another sentence that doesn't make sense.
23. Page 15 line 4 – replace "goes" with "is"
24. Page 15 line 7 – replace "available on this" with "available in this"
25. Page 15 line 10 – replace "This actually concerns" with "These are the"
26. Page 15 line 16 – replace "an" with "a"
27. Page 15 lines 21 and 14 – replace "centre" with "center"
28. Page 15 line 27 – replace "concerned" with "affected"
29. Page 15 line 29 – replace "estimates" with "estimate"
30. Page 16 line 2 – add "Then for all x > 0," before "the cdf can be computed as follows", and remove the $\forall x > 0$ from the equation
31. Page 16 line 7 – replace "based on a L-" with "based on an L-"
32. Page 17 line 6 – add the word "and" before "split-sample experiments"
33. Page 17 line 13 – replace "good reliability" with "reasonable reliability"
34. Page 18 line 2 – add the work "generally" before "included in the SCOPE Hydro"
35. Page 22 line 14 – reword to "highlights the fact that events covering more than 70% of France have only occurred after 1940".
36. Page 23 line 1 – replace "hit" with "affected"
37. Page 23 line 14 – replace "On the contrary" with "In contrast"
38. Page 23 line 15 – replace "largely" with "widely"
39. Page 24 figure 16 – replace "median of the number of days" with "median start date, as the number of days"
40. Page 26 line 3 – replace "excepted" with "except"
41. Page 26 lines 1, 3 and 18 – replace "hit" with "affected"
42. Page 26 lines 2 and 11 - replace "On the contrary" with "In contrast"
43. Page 26 line 3 – replace "excepted" to "except"
44. Page 26 line 5 – remove "on the contrary"
45. Page 26 line 10 – add apostrophe to 1893 and 1990 events' start dates
46. Page 27 line 21 – replace "on" with "of" in subheading

47. Page 27 line 27 – replace "largely documented" with "extensively documented"
48. Page 27 line 19 – replace "hit" with "affected"
49. Page 27 line 30 – replace "1975 on" with "1975 onwards"
50. Page 28 line 30 – replace "hit" with "affected"
51. Page 29 line 2 – replace "in a spatio-temporal view" with "from a larger scale spatio-temporal perspective"
52. Page 29 line 2 – replace "want to draw a more" with "want to make a more"
53. Page 29 line 4 – replace "matching of event" with "matching of events"
54. Page 29 line 6 – replace "consist in" with "consist of"
55. Page 29 line 7 – replace "this can lead to identify multiple" with "this can lead to identification of multiple"
56. Page 29 line 7 – add comma after "dataset" followed by "but"
57. Page 29 line 18 – replace "the right way forward" with "a promising way forward"
58. Page 30 lines 6 and 10 – you use GR6J-Cemaneige and then GR6J+Cemaneige – do you mean the same thing by these? If so be consistent.
59. Page 30 line 11 – replace the word "evolved" with a more appropriate word "expanded"? "decreased"?
60. Page 30 line 20 – place comma after hydrometeorological studies and remove "-"
61. Page 30 line 21 – remove "-" so comma follows "flows"
62. Page 30 line 21 replace "the limited historical depth of surface" with "the limited historical amount of surface"
63. Page 30 line 28 – replace "across a set of stations" with "space"
64. Page 30 line 32 – replace "qualified" with "quantified", or say "these events are qualified, quantified and compared …"
65. Page 31 line 2 – replace "improve our knowledge on historical events by taking advantage of the higher historical depth of upper-air atmospheric data" with "improving our knowledge of historical events by taking advantage of the more abundant historical upper-air atmospheric data"
66. Page 31 line 9 – replace "deeper" with "more comprehensive"
67. Page 31 line 10 – replace "proxys" with "proxies"
68. Page 31 line 20 – replace "match generally well the observed ones" with "generally match the observed ones well"
69. Page 31 line 27 – remove "in" in "used for in the spatial"

**C**

[Figure]

**P**

---

## Referee Comment (RC2) · T. Mathevet (Referee) · 15 Oct 2016

Review of the paper "Ensemble reconstruction of spatio-temporal extreme low-flow events in France since 1871", by Caillouet, Vidal, Sauquet, Devers & Graff. Review by Thibault Mathevet (15/10/2016)

Synthesis of my review :

Even if this paper appear to be rather long and sometimes "dense", I really appreciated reviewing this paper. I am very happy to congatulate authors for such an amount of work and very useful information and analyses on the drought history over France, since 140 years. Having an experience on data-rescue and long-term historical reconstructions, I consider that this work could have many applications, both in terms of reseach activities or operational hydrology. This work could also help hydrologists to

communicate with water managers, decision-makers or stakeholders, in order to show them exemples of long-term hydrological variability. I really hope that SCOPE hydro time-series would be available soon ?

I would rate the scientific significance and quality as Excellent. However, I rate the presentation quality as Fair to Good, because some paragraphs appear to be difficult to understand, even with carefull attention. I would like to invite authors to improve the explanation in a more pedagogical way of §2.2.2 (Bias correction and Schaake Shuffle) and 3.2.2 (spatial matching procedure, also used for the ensemble case). This could undermine our appreciation of the quality of the paper, even if §4 and §5 are very interesting.

It might not be the objective of the authors, but a paper in two parts could be easier to read, with a first part considering the methodology (basicaly from 20CR-SANDHY-SUB datasets to SCOPE climate) and a second part considering hydrological analyses and the discussion (basicaly, SCOPE Hydro and hydrological analyses).

Major comments :

§2.2.2 SCOPE Climate : this paragraph presenting the bias correction via a resampling-based correction approach and improvement of spatial coherence via Schaake Shuffle should be improved in order to be easily understood ;

§3.2.2 spatial matching procedure : the overlapping process is not clear. This paragraph should be improved in order to be easily understood

* the step from Fig 4a to Fig 4b is not clear on this example : I don't understand why two independent events are considered for red and grey colors, while there is only one event considered with the purple color ? Station 11 event definition should be continous during period covered by red and grey colors ?

* the step from Fig 4d to Fig 4e is not clear on this example : again, I don't understand why an event could be discontinuous, for the two blue and two green events ? p15, l7,

Fig 7 : again, I don't understand why there is only two spatio-temporal events and not four ?

§3.1 hydrological modeling : since the aim of this study is to represent particularly well drought events and that it is well-know that hydrological models are performing poorly on drought, why authors didn't consider an objective function based on hydrological signitures specific for drought, such as distribution of drought duration, severity, etc (VCN 10, VCN30, ...) ?

Minor comments :

p4, l26 : problem with the length of the line ;

p6, l11 : it might be out of the scope of this paper, but have you tryed to analyse the 20CR-SANDHY-SUB bias using a weather type classification (the seasonal classification is interesting but, beyond seasons weather type proportion might change from a season to another) ? ;

p7, l21 : KGE is expressed as KGE = 1-SQRT (...) ;

p7, §3.1 : a table with quantiles of catchments caracteristics and summary of performances (KGE, r, alpha, beta) might be interesting (as Table 2, in Pushpalatha et al., 2012);

p14, l12 : is the number of members to consider an event (10 on Fig 6 example) adapted from one station to another or roughly selected for the 662 stations ? If it's different from one station to another : give some quantile to precise the variability of this threshold ? Have you tested an unique value for the whole station sample ? ;

p17, l4, Fig 2 : I would appreciate to see distributions or boxplots of r, alpha, beta and KGE criteria ;

p17, §4 : again, it might be out of the scope of this paper, but it could be interesting to caracterise SCOPE hydro performances for drought simulation using hydrological

signatures and/or probabilistic criteria, such as CRPSS, etc. ? ;

p18, figure 10 & p19, figure 11 : for the ones not used to duration values and severity values, it could be interesting to put a panel on these figures with the distributions of event durations and severity obtained with the Observation or Safran Hydro. Another option would be to add a second y-axis with the quantiles corresponding to the duration/severity values ? ;

p22, fig 14 : what is the total spatial extent of the 622 hydrological stations ? what is the proportion of gauged surface over the France surface ? ;

p22 : It would be interesting to distinguish snow-dominated catchments and rain-dominated catchments and show a figures with the spatial extent of drought, given these two main processes (snow/rain)? ;

p25 & p26 l14-22 : given the length and density of your paper, Figure 17 and its related §do not appear necessary for me ;

p29, §6.4 : have you compared the Safran Hydro and SCOPE Hydro analyses on the 1958-2012 period, where hydrological simulations are both available ? A scatterplot of duration, severity or spatial extent by year could be interesting ? ;

p30, §6.6 : considering drought simulation, my experience is that conceptual RR models could be strongly biased. In a future work, you could consider a very simple method, using a bias correction of streamflow simulations by quantile classes, as proposed by F. Bourgin in its PhD at IRSTEA.
* * *

---

## Referee Comment (RC3) · Anonymous Referee #3 · 21 Oct 2016

General comments Overall I find this an excellent paper which makes an important contribution to understanding French low flows but also in providing a highly transferable method for reconstruction of daily river flows in other countries/locations. The authors are to be congratulated; this paper provides a benchmark study. The paper would however benefit from some additional work on grammar and tightening the communication. Some aspects are a little hard to follow and would benefit from some careful thought on how to present in a clearer way to help the reader follow what is going on. I think that a lot of the appendix information should be integrated – e.g. the work flow figure is very useful to understanding what is going on. All of my specific comments below are minor and try to be constructive. Finally I apologise from my delay!

Specific comments In section 2.2.2 SCOPE climate it would be useful if the authors could introduce the data, then outline the steps in its use and then deal systematically

with the additional treatments...for example it would be helpful to the reader of the final sentences in the section appeared earlier. These steps could then be used to organise the section.

Given the objective of creating an ensemble reconstruction, why was only one hydrological model used and why is no consideration given to the uncertainty in the GR6J parameters. The model is calibrated for the period Jan 1973- sept 2006. Was there a reason for choosing this period based on variation in flow conditions? I ask as the model is expected to reconstruct conditions that are potentially very different from the calibration period. If the focus is on low flows – was consideration given to how the model performed for different duration/intensity events during the period of observations.

I note that validation across the full set of catchments is not shown but is done – what were the salient points – it would be useful to summaries these in a couple of sentences.

"It has to be noted that thorough validation experiments not shown here – out-of-sample experiments, split-sample experiments – have been performed to carefully quantify the overall hydrological modelling performance."

I find the spatial mapping procedure difficult to follow and its communication would benefit from more clearly laying out the steps and then showing the example application.

Is it possible to make a conclusion around which aspect of drought – severity or duration – uncertainties are greatest?

When reporting seasonality you mention no visible trend – is there evidence of trend in the other parameters- severity/duration? There would seem to be for severity in the Correze catchment.

Please dont start section 4 with figures – text first.

On page 22 the text states "More generally, this figure highlights the fact that the only

events having hit more than 70% of France occurred after 1940." How confident can you be that this is a real trend or an artefact of the quality of the underlying data. While 20CR and such reanalysis data are hugely valuable confidence will reduce in time. Just a thought to consider which might be mentioned in the discussion.

I dont think there is a need to have so many sub sections in 6.2 – these would be better consolidated.

I think it would be more helpful to the reader to have the workflow image and other material in the appendices integrated into the text. This would not lengthen the paper and increase its readability.

The next generation 20CR gets back to 1850 if I am not mistaken. It would be useful to indicate this here with the potential to extend a further 20 years.

Minor points – note exhaustive - the paper needs a rigorous editing. Line 1: Consider a different phrase to 'historical depth' why not length? Line 7: drop the 'a' before continuous hydrological modelling.... Line 15: contribute to improving our knowledge of historical events.... Line 16: change to 'Moreover, this study allows for....' Perhaps the abstract might highlight that the methods presented are transferable to other locations where 20CR reconstructions are skilful. Page 2, line 3: drop 'in databses' Page 2, line 8: reword to have increased the breadth and depth of steamflow data over France. Would be useful to state why this work is insufficient. And to perhaps better clarify what you mean by breadth and depth. Page 2 Line 11 'have been employed' rather than have been developed. Page 3 line 8: has prompted Page 3 Line 9 of the 20th century rather than on and no need for at – just and using.... Page 3 Line 11 an analogue method Page 3 Line 14 combined with a physically based..... Page 3 Line 17 allow rather than allows Page 3 line 27 – final sentence of paragraph needs to be reworded Page 4 line 4 – delete- beforehand the refinement steps.... Page 4 line 27 – are these daily observation since gauge commencement? Page 5 line 4 – on an 8 km resolution grid...drop the and.. Page 5 line 6 – reword and notably for some of them

to some dedicated to low flows... Page 5 – line 14 – paving France? Page 6 – line 2 – double brackets needed for references Page 6 – line 3 – i dont follow how this is done – could the authors add more detail here. Page 6 line 8 – no comma after hypen. Would reword to necessitating a biad correction step. Page 6 line 28 – from the period rather than among the period – plus overuse of period in the sentence Page 6 line 35 - summarised by the 19 diagram in Annex B – please use better grammar Page 7 line 15 – no comma after hyphen Page 10 line 12 – delete 'The France subdivision in'... Page 10 line 17 many instead of much Page 10 line 10 matching rather than match Page 15 line 28 – the surface area of France Page 15 line 29 – This estimate Page 22 line 4 – Information not provided rather than an information not provided Page 23 line 5 – stations scattered in France? Page 26 line 3 – reword events both hit the entire France excepted the high-elevation snow-influenced alpine stations Page 27 line 24 reveal singular Page 27 line 25 – check for grammar Page 28 line 7 – reword - this can lead to identify multiple spatio-temporal events

---

## Author Comment (AC1) · 20 Dec 2016

*In this paper, the authors set out to reconstruct low-flow events in a reference network of catchments across France from 1871 to present. Due to the lack of observed streamflow records prior to the 1950s, downscaled climatological data (20CR) is used to reconstruct streamflow using hydrological models. Using knowledge of more recent low flow events over the past 50 years, the authors validate their novel approaches of low-flow event identification, spatial matching, and hydrological modelling, which succeed in identifying the well-known events of 1976 and 1989-1990. The approach highlights two additional extreme low-flow events in the 19th century: 1878 and 1893 and the authors conclude that many severe, long and widespread drought events occur prior to 1950.*

*This paper provides a valuable contribution to science in both methodology and results. The methods are many and varied and though multiple subjective decisions are included, they are well thought out and discussed. Whilst the many steps of the spatial matching procedure are quite challenging to decipher, the results are well presented. The combined threshold level is a particularly interesting concept. This is a very long paper, which might daunt the average reader, but the content is valuable and in my opinion the many maps and graphs all convey interesting information. The maps in Figures 15 and 16 are of particular merit, though Figure 15 would benefit from some slight adjustments (see comment below).*

The authors would like to thank Referee 1 for his positive comments on the manuscript. We also thank him/her for the specific and technical comments (in italic below) that will lead to improve the manuscript. The detailed answers to the specific comments are presented below.

*If the editor deems the paper too long (which would be understandable), there are several ways in which the main body of this manuscript could be cut down:*

*1. The introduction could be reduced.*

We believe the introduction as it is introduces the reader with the numerous different concepts covered in this paper. A shortened introduction might decrease the understanding of all facets of the topic. Nevertheless, we will check again if it can be shortened.

*2. The description of the derivation of the SCOPE climate data from 20CR-SANDHY-SUB could be stated much more succinctly in the main text, and a detailed description provided in the appendix.*

Thank you for the suggestion. We will indeed only present the SCOPE Climate dataset in this section and detail the entire SCOPE downscaling method (with more details) in the appendix.

*3. Similarly the description of the spatial matching procedure is lengthy, and could be summarised with a longer description in an appendix. Those who wish to reproduce your methods would be willing to read the appendices, whereas others, who are mainly interested in the results would not need to understand that level of detail.*

The aim of this paper is to focus on the methodology developed for identifying spatio-temporal events, a methodology that could be transposed in other contexts. In this way, the results are only shown as basic example results of the method. As a consequence, we would like to keep the entire method description in the main text.

*4. The day of the year analysis in figure 12 could be cut out, however it does show the spread of the start dates.*

As this figure is the only example of low-flow seasonality, we would prefer to keep it in the main text.

*5. Furthermore, Figure 13 could be removed, as the majority of the information is given in figures 10 and 11. Again however, it does show the linear relationship between severity and duration of events.*

Thank you for the suggestion. This figure will indeed be moved to the appendix or to some supplementary material.

*Comments that must be addressed:*

*1. The derivation of the SCOPE climate data is not well explained. Page 6 needs a lot of attention. It appears to be a significant amount of work that is not published elsewhere. This procedure needs to be clearly described, and moved to the appendix. For example, it is not clear what variables from the 20CR are downscaled (500hPa geopotential height?)*

We agree that this procedure would require a lot more details to be fully understandable, and we also agree that having a deeper description in the appendix would be the way forward. As mentioned above, only a description of the SCOPE Climate dataset would be kept in the main text. Considering the question above: 20CR output variables are only used as predictors (and not downscaled). As detailed in Caillouet et al. (2016), 20CR large-scale predictors (geopotential height, air temperature, humidity and vertical velocity at different pressure levels, as well as sea surface temperature) are used to derive local-scale predictands (precipitation and temperature) in the SANDHY-SUB statistical downscaling method. SCOPE adds two more steps (which will be detailed in the appendix) to SANDHY-SUB.

*2. Please indicate the ideal value of KGE (1 I assume?) on both the description of the metric on page 7 and the map given as Figure 2. Furthermore, please check the equation provided on page 7, as this indicates that a low value should be a good score. Gupta (2009) provide the equation you present as the Euclidian Distance (ED), whilst KGE is 1-ED.*

Indeed, the equation is incorrect. This will be corrected. The ideal value of KGE (1) will also be specified.

*3. Your description of the spatial matching procedure needs some work to improve its clarity. I suggest it is lengthened and sent to an appendix.*

As the wish of the authors is to produce a methodological paper with some example results, we want to keep the spatial matching procedure in the main text. We will try to work on the clarity of the method.

*4. Page 25 Figure 17 – add a scale/legend to this figure*

Indeed, it has been cut, this will be corrected.

*5. Page 31 lines 3 and 4 – You state "Result also highlight that the worst events in terms of severity and duration or spatial extent often belong to the pre-1950 period". This statement is not backed up by your graphics. Figure 14 indicates the majority of the most widespread events were post 1940, Figure 15 indicates that 1990 and 1976 were the longest, and similarly in figure 13 only 1945 is picked out among the most severe in the Corrèze catchment. Furthermore, figures 10 and 11 do indicate some long and severe droughts prior to 1950, but the latter period on these graphs seem to show significantly more events and more serious events than the earlier period. I think what you mean to say is that many severe, long and widespread events occur prior to the 1950s, but I don't think it's true that the majority of the most severe events are in the early part of your reconstructions. Please amend this sentence accordingly.*

Indeed, this is a mistake in the sentence writing. This is not pre-1950 but actually post-1940. This will be changed accordingly.

*Further recommended minor adjustments and comments:*

*1. Page 2 lines 2 to 6 – references here may be assumed to be based in French catchments due to the outlining of the lack of available data in France, then using the word consequently to start the next sentence. Rephrase to indicate these are not all French studies.*

It will be clarified.

*2. Page 3 lines 3 and 4 – "high spatial and temporal scales" and "large-scale atmospheric and oceanic data" high scale means different things to different people. Do you mean high resolution or not?*

Indeed, this is a mistake. The right formulation is "small spatial and temporal scales", this will be corrected.

*3. Page 3 line 13 – within which country was Dayon's study based?*

Dayon's study is based within France, this will be specified.

*4. Page 3 line 16 – downscaling of 20CR – what variables did you downscale and at what resolution?*

As mentioned previously, this will be added in the details of the SCOPE method in appendix. Large-scale (2.5°) predictors are (1) temperature at 925 hPa and 600 hPa, (2) geopotential height at 1000 hPa and 500 hPa, (3) vertical velocity at 850 hPa and (4) humidity as a combination of the relative humidity at 850 hPa and precipitable water content in the entire column.

*5. Page 3 line 34 to page 4 line 1 – you mention the use of probabilistic data to account for uncertainty, but very few of your figures display the uncertainty from the ensemble in the results (only really Figures 10 and 11). It would be nice to see some spatial mapping of uncertainty – perhaps a map of one of your events, with density of colour according to how many of the 25 ensemble members identify that event, this map could accompany figure 17.*

Thank you for this suggestion, this would indeed be a nice figure. For a specific event, the spatial pattern would be similar to the one obtain for duration or severity. Indeed, there is a higher probability that a high number of members detects the event if the latter is particularly long or severe for a station. Below is an example of the number of members detecting the 1893 event (to be compared to corresponding maps of return period in duration and severity in Fig. 16):

[Figure]

As this paper is a methodological paper with already a lot of example results, this will not be added. But a sentence such as: "The more severe or the longer the event, the higher the number of members detecting the event." will be added.

*6. Page 4 line 19 – what does "without direct human influence" mean?!*

This means without abstractions, derivations and reservoir operations, i.e. human influence of catchment processes

*7. Page 4 line 27 – we are referred to Annex A which mentions Safran Hydro and SCOPE Hydro before the datasets are introduced. I suggest you add "see section 2.2.1 and 2.2.2" to the caption of Figure 18.*

This will be added.

*8. Page 7 line 20 – I don't think it can be said to give "equal weights", better to say "reduces the bias towards high flows"*

This will be corrected.

*9. Page 7 line 24 – how many stations used CemaNeige then? State this. Could you highlight them somehow on Figure 2?*

All stations used for CemaNeige. CemaNeige is calibrated locally on 187 stations with high snow influence. Median values of the calibrated parameters are then used for the simulations on all 475 remaining stations. This will be rephrased.

*10. Page 7 line 26 – so all the catchments were given the same values for the 2 CemaNeige parameters? Is this realistic? Or were the catchments calibrated several times, and then the median of those given to each individual catchment? This needs to be clearer.*

Only catchments with little snow influence (475 catchments) have the same CemaNeige parameters. Using fixed parameters for these catchments allows keeping realistic values as there were not enough snowfall episodes during the calibration period to obtain realistic calibrated CemaNeige parameters. This will be rephrased.

*11. Page 8 lines 17-24 – I appreciate that this algorithm comes from reservoir design, but it would be more appropriate to explain it in the context of flow deficit here.*

This will be adapted to this context.

*12. Page 8 line 25 - Your end date is defined as the time of maximum depletion. Do you think this is representative of the end of an extreme low flow/drought period? What about the time it takes for the stream to return to "normal" flows? See Parry et al (2016)*

*http://ppg.sagepub.com/content/early/2016/06/02/0309133316652801.refs*

Our end date corresponds to the maximum depletion, so actually to the return to the normal flow, i. e to the date where the flow exceeds the threshold again (but not the normal deficit). But indeed, the date corresponding to the return to 0 deficit would be useful for assessing drought recovery. This would be a nice additional study which is out of the scope of this paper.

*13. Page 11 Figure 4 – Station 11 shows a single black bar in (a), but a red and a grey bar in (b), is this actually one event or is fig (a) not quite clear enough?*

In fact, these are two very close but independent events with only a few days between them, hence the two colors. The resolution of the figure will be increased in order to clearly distinguish the two separate events.

*14. Page 14 Figure 7 – why is the late 1990 event not picked up by Safran?*

This event is not picked up for this particular station but is actually picked up for all other stations of the HER (see Discussion section 6.3, p. 28, l.27-30). This may happen locally, due to the sensitivity of the method to different parameters like the local threshold.

*15. Page 14 Figure 7 – this figure takes a very long time to load on my PC – it interrupts scrolling significantly, and almost crashes my browser. Is its file size much larger than Figure 4? If so why? Can it be reduced without compromising its quality?*

We actually had the same problem (even if the file size was the same than for the other figures). The figure has been created differently and it now works correctly.

*16. Captions for Figures 8 and 9 – don't think you need to point out the different scales on the yaxes.*

Thank you for the suggestion.

*17. Page 16 – provide an assessment of the bias in SCOPE Hydro (median) compared with observed flows, for low and high flow seasons. Calculate the percentage of time the obs are within the SCOPE Hydro range for all stations.*

SCOPE Climate and SCOPE Hydro will be made available in forthcoming data papers. They will also provide a deeper description of these two datasets, including median bias, KGE or reliability. As this paper is already very dense, we do not wish to add further validation results. For information, the maps of the KGE decomposition (in median) are the following (alpha as variance, bias as beta and r as linear correlation on the calibration period):

[Figure]

18. *Page 17 line 12 – not sure I agree that the observations are "most of the time" included in the SCOPE Hydro range – the obs seem to be on the periphery most of the time – address with comment above.*

This will be rephrased. Further analyses on the reliability of the SCOPE Hydro ensemble (that will not be shown here) actually support this statement.

19. *Page 18 Figure 10 and page 20 Figure 12 – the dashed lines are visible on the computer, but not on the print out for me. This could easily be a problem my end, but please double check it prints correctly for you.*

We did not encounter this problem.

20. *Page 18 figure 10 – the "Whiskers" (note spelling!) do not extend to 1.5 times the IQR, they extend to the largest (and smallest) observations still within 1.5 times the IQR*

Indeed, this will be corrected.

*21. Page 19 line 17 – "A higher number of extreme events and higher severity values are simulated after 1940 for the Corrèze" – do you have any stats to prove this? Mann-Whitney U test for step change? Is there statistical significance?*

These assumptions are only based on the figures and detailed statistic trend tests have not been considered as the aim of the paper was to provide basic examples of the method.

*22. Page 20 line 9 – "There is no visible trend on the seasonality of start dates" – did you do any statistical tests?*

Cf. answer to above comment.

*23. Page 22 Figure 14 – You have displayed months and years on this plot whilst all other plots just categorise by year. You could remove the months from this plot to make the plot clearer.*

Other plots are categorised by event names (actually a year) on the contrary to this plot which is categorised by date of maximum spatial extent. Keeping the months allows a better understanding of the seasonality of the event and a coherence with the x-axis. Moreover, removing the month might confuse the reader as the names of the events do not always correspond to the year of maximum spatial extent (for example the 1990 event corresponds to 12/1989).

*24. Page 23 Figure 15 – this is a great Figure, but it really isn't colour-blind friendly. Remember 1 in 10 men is colour-blind. I know this will be difficult to take into account with a map like this, but I suggest you make sure that the 3 or 4 main events you are picking out are in contrasting colours that colour-blind people can differentiate from the others. At the moment, for those with Protanope colour-blindness (the most common) 1990 is clearly visible, 1893 and 1985 look the same as each other, 1878 and 1978 look the same as each other, 1943 and 1949 look the same as each other, and the rest are all very similar. Try using the mobile app "CVSimulator" to test your images (I've run your image through this app, see graphic at end of document). On this note – Figures and 4 and 7 use orange and green which are indistinguishable to the colour-blind. Reconsider this colouring if possible. Figures 16 and 17 are OK, Figure 20 isn't but I don't think the issue can be avoided here sadly.*

Thank you for this remark and this very interesting app. We tried to use colorblind safe color scales even if the result is not always perfect. Figure 15 requires 12 different colours and it is not possible to find an entire adapted color scale. Nevertheless, we will indeed try and change the colors to distinguish the major events from each other.

*25. Page 23 line 4 – I think you have 1878 and 1893 the wrong way round in this sentence*

You are right, this will be corrected.

*26. Page 23 lines 7 to 9 – comment on the 2003 event – this is a major finding, I suggest you add this to the conclusion!*

This result should be taken carefully. As mentioned, it is not possible to automatically compare events from Safran Hydro and SCOPE Hydro but it is possible to compare exceptional events by manually selecting the events in each dataset. A comparison showed that the 1976 and 1990 events are very well reconstructed by SCOPE Hydro but the duration and severity of the 2003 event are understimated by SCOPE Hydro. This may be due to the lack of soil-atmosphere retroactions which are important for this event and not taken into account into our hydrometeorological reconstruction chain. A sentence will be added to the paper to prevent the reader from any over-interpretation.

*27. Page 26 lines 4 to 9 – this discussion seemed to suggest that 1990 wasn't a particularly severe event, whereas Figure 16 indicates it really was just as severe as the 1893 and 1976 events, especially in the massif central regions.*

The sentence should be rephrased with the removal of "on the contrary". The discussion focused on the fact that this event was exceptionally long for a majority of France, more than exceptionally severe.

*28. Page 26 lines 10 and 11 – you state that the 1990 event start dates lie within a common 6 month period for all stations concerned – however I can definitely see some green dots on that lower right map in Figure 16! I suggest rewording "all stations concerned" as "for the vast majority of stations"*

This will be corrected.

*29. Page 30 section 6.6 – no mention of parameter uncertainty and model structural uncertainty – e.g. GR4J and GR5J.*

Model structural uncertainty is mentioned with the use of the Isba-Modcou hydrological model. A sentence will be added on the parameter uncertainty.

*30. Page 30 conclusion – no mention of the use of Safran Hydro*

Indeed, the aim was to mention only the main datasets and results in order to make the conclusion more understandable.

*Spelling/Grammatical Errors*

Thank you for your very attentive reading, all your corrections will be taken into account.

References

Caillouet, L., Vidal, J.-P., Sauquet, E., and Graff, B.: Probabilistic precipitation and temperature downscaling of the Twentieth Century Reanalysis over France, Clim. Past, 12, 635-662, doi:10.5194/cp-12-635-2016, 2016.

---

## Author Comment (AC2) · 20 Dec 2016

*Synthesis of my review :*

*Even if this paper appear to be rather long and sometimes "dense", I really appreciated reviewing this paper. I am very happy to congatulate authors for such an amount of work and very useful information and analyses on the drought history over France, since 140 years. Having an experience on data-rescue and long-term historical reconstructions, I consider that this work could have many applications, both in terms of reseach activities or operational hydrology. This work could also help hydrologists to communicate with water managers, decision-makers or stakeholders, in order to show them exemples of long-term hydrological variability.*

The authors would like to thank Referee 2 for his positive comments on the manuscript and the specific and technical comments (in italic below) that will lead to improve the manuscript. The detailed answers to the specific comments are presented below.

*I really hope that SCOPE hydro time-series would be available soon ?*

SCOPE Climate and SCOPE Hydro will be made available as soon as possible in forthcoming data papers. In the meantime, preliminary packed datasets are available upon request to the authors.

*I would rate the scientific significance and quality as Excellent. However, I rate the presentation quality as Fair to Good, because some paragraphs appear to be difficult to understand, even with carefull attention. I would like to invite authors to improve the explanation in a more pedagogical way of $\S$2.2.2 (Bias correction and Schaake Shuffle) and 3.2.2 (spatial matching procedure, also used for the ensemble case). This could undermine our appreciation of the quality of the paper, even if $\S$4 and $\S$5 are very Interesting.*

Following our responses to comment from referee #1, we will only present the SCOPE Climate dataset in section 2.2.2 and describe the entire SCOPE method (with more details) in the appendix. Efforts will be made to improve the understanding of section 3.2.2. The issue may however come from the low resolution of Figure 4 which can lead to a confusion of event matching.

*It might not be the objective of the authors, but a paper in two parts could be easier to read, with a first part considering the methodology (basicaly from 20CR-SANDHY-SUB datasets to SCOPE climate) and a second part considering hydrological analyses and the discussion (basicaly, SCOPE Hydro and hydrological analyses).*

Writing up a two-part paper with this material would be quite difficult. See the responses to comments above on restructuring the climate part.

*Major comments :*

*§2.2.2 SCOPE Climate : this paragraph presenting the bias correction via a resampling-based correction approach and improvement of spatial coherence via Schaake Shuffle should be improved in order to be easily understood ;*

Indeed, SCOPE will be detailed in the appendix (see previous answer).

*§3.2.2 spatial matching procedure : the overlapping process is not clear. This paragraph should be improved in order to be easily understood*

*\* the step from Fig 4a to Fig 4b is not clear on this example : I don't understand why two independent events are considered for red and grey colors, while there is only one event considered with the purple color ?  Station 11 event definition should be continous during period covered by red and grey colors ?*

This come from the lack of resolution of Figure 4. The correct figure is the following:

[Figure]

Grey and red events are indeed two different events in 4(a), which was not clear in the version included in the manuscript.

*\* the step from Fig 4d to Fig 4e is not clear on this example : again, I don't understand why an event could be discontinuous, for the two blue and two green events ?*

4d to 4e is only a local report of the inter-HER matching to the stations of the HER. After the inter-HER matching, the first two events of HER 11 are matched together (two bars with the same blue color). If you go back to 4b, the local events corresponding to these two blue bars are the red and grey events. So after the inter-HER matching, these two independent events are matched together, giving the blue color instead of the grey and red. But this misunderstanding may also come from the same problem than previously (it was not possible to distinguish the two independent event for the grey and red bars).

*Fig 7 : again, I don't understand why there is only two spatio-temporal events and not four ?*

This figure only illustrates the report of the spatial matching from average events to SCOPE Hydro events. The colors you can observe in 7a are already the result of the spatial matching on average events (spatial matching which is not shown before, as Figure 4 corresponds to Safran Hydro and not average events). So the spatial matching on average events gives two different events: one orange and one green. 7b shows the report of these two events to the SCOPE Hydro events. At the top of the figure, Safran Hydro events are only shown for information as it is used in the discussion (this is the only line which refers to a result shown in Figure 4). Maybe it would be clearer to choose two different colors for average events and events of the 25 members (than orange and green) as it is completely independent of Figure 4.

*§3.1 hydrological modeling : since the aim of this study is to represent particularly well drought events and that it is well-know that hydrological models are performing poorly on drought, why authors didn't consider an objective function based on hydrological signitures specific for drought, such as distribution of drought duration, severity, etc (VCN 10, VCN30, …) ?*

SCOPE Hydro has been created in order to be used in any type of hydrological studies. For this reason, the objective function has been chosen to give equal weights to high and low flows.

*Minor comments :*

*p4, l26 : problem with the length of the line ;*

This will be corrected with the final version.

*p6, l11 : it might be out of the scope of this paper, but have you tryed to analyse the 20CR-SANDHY-SUB bias using a weather type classification (the seasonal classification is interesting but, beyond seasons weather type proportion might change from a season to another) ? ;*

We didn't try but indeed, it is a nice suggestion to better understand the origin of biases.

*p7, l21 : KGE is expressed as KGE = 1-SQRT (…) ;*

Indeed, this will be corrected.

*p7, §3.1 : a table with quantiles of catchments caracteristics and summary of performances (KGE, r, alpha, beta) might be interesting (as Table 2, in Pushpalatha et al., 2012);*

It might indeed be interesting and we will include it in the forthcoming paper describing SCOPE Hydro in detail. Please note that KGE values are available in the manuscript in Fig. 2.

*p14, l12 : is the number of members to consider an event (10 on Fig 6 example) adapted from one station to another or roughly selected for the 662 stations ? If it's different from one station to another : give some quantile to precise the variability of this threshold ? Have you tested an unique value for the whole station sample ? ;*

This number is different from one station to another. Below is the map of the final values:

[Figure]

A unique value has not been tested as having a different value for each station allows a re-calibration against Safran to improve low-flow event identification. But except for specific stations, values are kept between 8 and 12.

*p17, l4, Fig 2 : I would appreciate to see distributions or boxplots of r, alpha, beta and KGE criteria ;*

See our response to a comment above. The manuscript is already very long and dense and we would rather not add more figures not directly related to extreme low-flow events.

*p17, §4 : again, it might be out of the scope of this paper, but it could be interesting to caracterise SCOPE hydro performances for drought simulation using hydrological signatures and/or probabilistic criteria, such as CRPSS, etc. ? ;*

Cf. Response to comment above, and response to referee #1 for a figure showing the median performance of SCOPE Hydro and Safran Hydro in terms of KGE and KGE decomposition (alpha for variance, beta for bias and r for linear correlation on the calibration period).

[Figure]

*p18, figure 10 & p19, figure 11 : for the ones not used to duration values and severity values, it could be interesting to put a panel on these figures with the distributions of event durations and severity obtained with the Observation or Safran Hydro. Another option would be to add a second y-axis with the quantiles corresponding to the duration/severity values ? ;*

As the information brought by figures is already very dense, we would be prefer not to add such a panel. However, the idea of translating values in mm /days into long-term quantiles would be an interesting way of presenting the results, and we will keep it in mind for further analyses.

*p22, fig 14 : what is the total spatial extent of the 622 hydrological stations ? what is the proportion of gauged surface over the France surface ? ;*

The gauges surface corresponds to around 41% of the France surface.

*p22 : It would be interesting to distinguish snow-dominated catchments and raindominated catchments and show a figures with the spatial extent of drought, given these two main processes (snow/rain)? ;*

This would be a very nice extension of this paper. As this paper is mainly a methodological paper, we do not which to extend the results.

*p25 & p26 l14-22 : given the length and density of your paper, Figure 17 and its related §do not appear necessary for me ;*

Figure 17 is the only figure providing an ensemble characterisation of a spatio-temporal extreme low-flow event. Figure 13 will be removed as it is partly redundant with figures 10 and 11.

*p29, §6.4 : have you compared the Safran Hydro and SCOPE Hydro analyses on the 1958-2012 period, where hydrological simulations are both available ? A scatterplot of duration, severity or spatial extent by year could be interesting ? ;*

We actually did this analysis manually for a few exceptional events. Drawing a scatterplot as suggested would require a formal temporal comparison of spatio-temporal extreme low-flow events across different datasets (which is difficult for now -- see section 6.4). To give a concrete example, the x-axis of Fig. 10 and Fig. 11 corresponds to the name of the events, itself corresponding to a spatial center date, independent for each dataset. If we do not link events across datasets, two events occurring during the same period (and that should be linked together) will be identified differently in the x-axis. This would most likely generate more questions than answers and this could hardly be done in an automated way.

*p30, §6.6 : considering drought simulation, my experience is that conceptual RR models could be strongly biased. In a future work, you could consider a very simple method, using a bias correction of streamflow simulations by quantile classes, as proposed by F. Bourgin in its PhD at IRSTEA.*

As shown in the response to referee #1 and more specifically the figure plotting biases between SCOPE Hydro and Safran Hydro, a slightly negative bias may be detected, but remains largely under 10%. Moreover, we didn't want to implement any streamflow quantile-quantile bias correction as it would add some more temporal transferability hypotheses in the hydrometeorological modeling chain.

---

## Author Comment (AC3) · 20 Dec 2016

*Synthesis of my review :*

*General comments Overall I find this an excellent paper which makes an important contribution to understanding French low flows but also in providing a highly transferable method for reconstruction of daily river flows in other countries/locations. The authors are to be congratulated; this paper provides a benchmark study. The paper would however benefit from some additional work on grammar and tightening the communication. Some aspects are a little hard to follow and would benefit from some careful thought on how to present in a clearer way to help the reader follow what is going on. I think that a lot of the appendix information should be integrated – e.g. the work flow figure is very useful to understanding what is going on. All of my specific comments below are minor and try to be constructive. Finally I apologise from my delay!*

The authors would like to thank Referee 3 for his positive comments on the manuscript. The detailed answers to the specific comments (in italic) are presented below.

*Specific comments In section 2.2.2 SCOPE climate it would be useful if the authors could introduce the data, then outline the steps in its use and then deal systematically with the additional treatments...for example it would be helpful to the reader of the final sentences in the section appeared earlier. These steps could then be used to organise the section*

For a better understanding, we will detail the SCOPE method in appendix. Only the SCOPE Climate dataset will be detailed in the main text. See also the related responses to referee #1 and referee #2.

*Given the objective of creating an ensemble reconstruction, why was only one hydrological model used and why is no consideration given to the uncertainty in the GR6J parameters. The model is calibrated for the period Jan 1973- sept 2006. Was there a reason for choosing this period based on variation in flow conditions? I ask as the model is expected to reconstruct conditions that are potentially very different from the calibration period. If the focus is on low flows – was consideration given to how the model performed for different duration/intensity events during the period of observations.*

There is a work in progress concerning the hydrological modeling structural uncertainty (see section 6.6). This will be presented in a future paper. The calibration period has been chosen as 90% of the observed date were available during this period. Moreover, it indeed includes three important extreme low-flow events (1976, 1990, 2003).

*I note that validation across the full set of catchments is not shown but is done – what were the salient points – it would be useful to summaries these in a couple of sentences. "It has to be noted that thorough validation experiments not shown here – out-of-sample experiments, split-sample experiments – have been performed to carefully quantify the overall hydrological modelling performance."*

Thank you for this suggestion. A few salient points will be added to the manuscript.

*I find the spatial mapping procedure difficult to follow and its communication would benefit from more clearly laying out the steps and then showing the example application*

The principle is first explained (simply an overlap of dates). Then the definition of the required parameters is developed (spatial domain for the matching). Finally, the example only provides a better understanding of the use of two spatial domains (by HER and then over France). We do not think it would be possible to explain these two steps without an example. Nevertheless, we will try to improve the understanding of this step.

*Is it possible to make a conclusion around which aspect of drought – severity or duration – uncertainties are greatest?*

It would be difficult to establish such a statement as severity and duration are two different variables with different units (mm and days). It would be possible to compute the range of values using return periods but it would add uncertainties related to the distribution fitting. Moreover, these two aspects highly depend on the specific event considered. Figure 16 shows for example that there are more stations detecting a long event rather than a severe event in 1990. This is the contrary for the 1893 event.

*When reporting seasonality you mention no visible trend – is there evidence of trend in the other parameters- severity/duration? There would seem to be for severity in the Correze catchment.*

These assumptions are only based on the figures and formal trend tests have not been considered as the aim of the paper was to provide basic examples of the method.

*Please dont start section 4 with figures – text first.*

This will be corrected

*On page 22 the text states "More generally, this figure highlights the fact that the only events having hit more than 70% of France occurred after 1940." How confident can you be that this is a real trend or an artefact of the quality of the underlying data. While 20CR and such reanalysis data are hugely valuable confidence will reduce in time. Just a thought to consider which might be mentioned in the discussion.*

As mentioned previously, no statistical tests have been performed. We will mention uncertainties deriving from 20CR in the discussion (already discussed in Caillouet et al., 2016).

*I dont think there is a need to have so many sub sections in 6.2 – these would be better consolidated.*

Thank you for the suggestion. As this paper covers many subjects, the subsections allow the reader to quickly find a specific item.

*I think it would be more helpful to the reader to have the workflow image and other material in the appendices integrated into the text. This would not lengthen the paper and increase its readability.*

Indeed, thank you for the suggestion, we will integrate this figure into the main text.

*The next generation 20CR gets back to 1850 if I am not mistaken. It would be useful to indicate this here with the potential to extend a further 20 years.*

You're right (even if there is less confidence in 20CR before the 1870s)!

*Minor points*

Thank you for your very attentive reading, your corrections will be taken into account.

*Page 4 line 27 – are these daily observation since gauge commencement?*

Yes, they are.

References

Caillouet, L., Vidal, J.-P., Sauquet, E., and Graff, B.: Probabilistic precipitation and temperature downscaling of the Twentieth Century Reanalysis over France, Clim. Past, 12, 635-662, doi:10.5194/cp-12-635-2016, 2016.

---

## Author Response (AR2)

Dear Kerstin Stahl,

Please, find attached the revised manuscript.
All the comments of the second referee have been taken into account.

Best regards,
Laurie Caillouet